# CD13 orients the apical-basal polarity axis necessary for lumen formation

Li-Ting Wang[1,2], Abira Rajah[3], Claire M. Brown[3,4] & Luke McCaffrey [1,2,5,6 ✉]

Polarized epithelial cells can organize into complex structures with a characteristic central lumen. Lumen formation requires that cells coordinately orient their polarity axis so that the basolateral domain is on the outside and apical domain inside epithelial structures. Here we show that the transmembrane aminopeptidase, CD13, is a key determinant of epithelial polarity orientation. CD13 localizes to the apical membrane and associates with an apical complex with Par6. CD13-deficient cells display inverted polarity in which apical proteins are retained on the outer cell periphery and fail to accumulate at an intercellular apical initiation site. Here we show that CD13 is required to couple apical protein cargo to Rab11-endosomes and for capture of endosomes at the apical initiation site. This role in polarity utilizes the short intracellular domain but is independent of CD13 peptidase activity.

[1] Rosalind and Morris Goodman Cancer Institute, McGill University, Montreal, Canada. [2] Division of Experimental Medicine, McGill University, Montreal, Canada. [3] Department of Physiology, McGill University, Montreal, Canada. [4] Advanced BioImaging Facility (ABIF) McGill University, Montreal, Canada. [5] Gerald Bronfman Department of Oncology, McGill University, Montreal, Canada. [6] Department of Biochemistry, McGill University, Montreal, Canada. ✉email: luke.mccaffrey@mcgill.ca

Epithelial cells are major building blocks for numerous tissues and organs that provide a barrier between tissue compartments and the external environment[1]. An important property of many epithelial cells is establishing polarity along an apical-basal axis such that the apical membrane faces a lumen that is contiguous with the external environment. This creates a barrier that allows selective vectoral transport of macromolecules for absorption or secretion. In addition, polarized epithelial cells spatially regulate signaling pathways that control diverse cellular properties including stem cell renewal, differentiation, survival, proliferation, metabolism, motility, and adhesion that control tissue growth and organization[2]. Disrupted apical-basal cell polarity and associated signaling pathways is frequent in epithelial malignancies, which account for greater than 80% of human cancers[3].

Epithelial cells are characterized by the presence of multiple dynamic complexes that associate with unique apical and basolateral membrane domains that are separated by tight junctions[1]. The majority of established regulators of apical-basal polarity are multi-domain scaffold and adaptor proteins, that associate with transmembrane proteins that anchor them to the plasma membrane[4]. For example, Crb3 is a transmembrane apical protein that interacts with Pals1, Patj, and Par6 to form an apical Crumbs complex[5]. Podocalyxin is another apical transmembrane protein that associates with Ezrin, which binds filamentous actin and links the apical membrane to the cytoskeleton[6]. Meanwhile, transmembrane junctional proteins bind to Par3 as part of an apicolateral Par-complex associated with Par6 and aPKC[7]. Therefore, Par6 is a multi-functional polarity adaptor that can associate with multiple polarity complexes through protein-protein interaction sites including a PDZ domain that binds Pals1 or Par3, a semi-CRIB domain that binds Cdc42, and a PB1 domain forms a heterodimeric PB1-PB1 interaction with aPKC.

Epithelial cells cultured in 3D basement membrane extract rich in laminin is a well-established model of apical-basal polarity[8,9]. In this model, apical and basolateral proteins are initially co-distributed on the plasma membrane of single cells. Following the first cell division, an apical membrane initiation site (AMIS) is established at the location of the cytokinetic midbody[10]. Rab-positive recycling endosomes have been identified as important players of apical trafficking[11]. Rab-dependent endocytosis internalizes apical membrane proteins and delivers them to the nascent apical domain at the middle of cell clusters, thus establishing an apical-basal polarity axis with an internal apical membrane that expands to establish a central lumen. Rab11 is an apically located small GTP-binding protein that plays a central role in transcytosis and recycling of apical proteins for lumen formation in diverse epithelial systems[12–17]. For example, Rab11 and the exocyst complex are required for lumen formation by delivering Crb to the apical surface via the Golgi and then apical recycling endosomes (ARE)[18,19].

Strikingly, depletion of mammalian polarity proteins often results in multi-lumen phenotypes resulting from misoriented cell divisions[20–22]. However, the apical-basal polarity axis remains intact with the apical domain oriented towards the interior and the basolateral domain oriented to the periphery of multicellular structures. Notably, inverted (inside-out) polarity is observed in some developmental contexts (e.g., preimplantation embryo) and is associated with metastasis of colorectal cancer cells[23]. However, the mechanisms that control the orientation of apical-basal polarity remain to be fully elucidated.

CD13 (Aminopeptidase N, APN) is a type II membrane-bound zinc-dependent metalloprotease that is widely expressed on surface of epithelial cells, immune cells, and fibroblasts. It can also exist as a soluble form in plasma, serum and urine[24]. CD13 has a short amino-terminal intracellular domain, a helical transmembrane anchor, an extracellular stalk that connects the C-terminal catalytic ectodomain. The peptidolytic ectodomain cleaves terminal amino acids from of peptides in diverse physiological processes including angiotensin activation, amino acid metabolism, ECM degradation, neuropeptide processing, and trimming peptides bound to MHC-II complexes[25–27]. In contrast, several studies have indicated that CD13 also functions independent of enzymatic activity, including cell-cell adhesion between inflammatory cells, intracellular trafficking in monocytes, and migration of fibroblasts and immune cells[27–30]. Many non-catalytic functions instead require a tyrosine (Y6) in the highly conserved intracellular domain. Y6 can be phosphorylated in a Src-dependent manner and mutation to phenylalanine abrogates cell adhesion[31,32] and β1-integrin recycling during cell migration[33]. Other non-catalytic functions for CD13 include acting as a receptor for coronaviruses, and it has been used as a receptor for tumor-homing peptides that guide experimental anti-cancer drugs to tumors expressing high levels of CD13[34,35]. In epithelial cells, CD13 localizes to the apical membrane, however its function in this cell type has largely remained elusive.

In this work we characterized a role for CD13 in controlling the orientation of epithelial cell polarity required for lumen formation by promoting the association and recruitment of apical polarity proteins to endosomes and delivery to the emerging apical domain. Depletion of CD13 expression impaired the endocytic trafficking of apical components, resulting in an inverted apical-basal polarity axis.

## Results

**CD13 associates with Par6 at the apical membrane.** CD13 was previously reported to localize at the apical membrane in epithelia and its expression increases during apical-basal polarization in Caco-2 cells[36,37], an established cellular model to study apical-basal polarity[20,22]. We therefore hypothesized that CD13 may have a role in apical-basal polarity. We first established that endogenous CD13 and exogenous CD13-V5 were enriched at the luminal membrane of 3D Caco-2 cysts and they colocalized with Par6 and other apical markers (aPKC, Ezrin, Pals1, F-actin) (Fig. 1a, b; Supplementary Figs. 1a–d). CD13 did not overlap with E-cadherin or ZO1, indicating that it does not localize to adherence or tight junctions (Supplementary Figs. 1e, f).

To determine if CD13 formed a complex with Par6 and/or aPKC, we performed co-immunoprecipitation experiments with V5-tagged CD13. These experiments indicate that CD13 associates with Par6 and aPKC in HEK293 (Fig. 1c) and Caco-2 (Fig. 1d, Supplementary Fig. 1g) lysates. Co-immunoprecipitation of endogenous CD13 and Par6 in Caco-2 lysates confirm the interaction (Supplementary Fig. 1h). Par6 can associate with either Par3 or Pals1 as part of the Par or Crumbs complexes respectively[38,39]. We detected Pals1 co-precipitating with CD13, whereas we did not detect Par3 in CD13 co-immunoprecipitates from HEK293 cells (Supplementary Fig. 1g). This indicates that CD13 interacts with Par6 as part of the Crumbs complex and is independent of the Par complex (Fig. 1d, Supplementary Fig. 1g). We further confirmed association of CD13 with Crumbs3 by co-precipitation experiments (Supplementary Fig. 1i).

We next wondered if CD13 was required for the association of Par6 with the Crumbs complex. We therefore knocked down CD13 using two independent shRNA and immunoprecipitated Par6. These data revealed that the association of Par6 with Pals1 was reduced by CD13 knockdown, whereas the association between Par6 and aPKC or Par6 and Par3 were not affected (Fig. 1e). These data provide further support that CD13 associates with Par6 and the Crumbs complex but not Par3. This is consistent with our localization data showing CD13 enriched in the apical membrane, but not tight junctions that contain Par3[40].

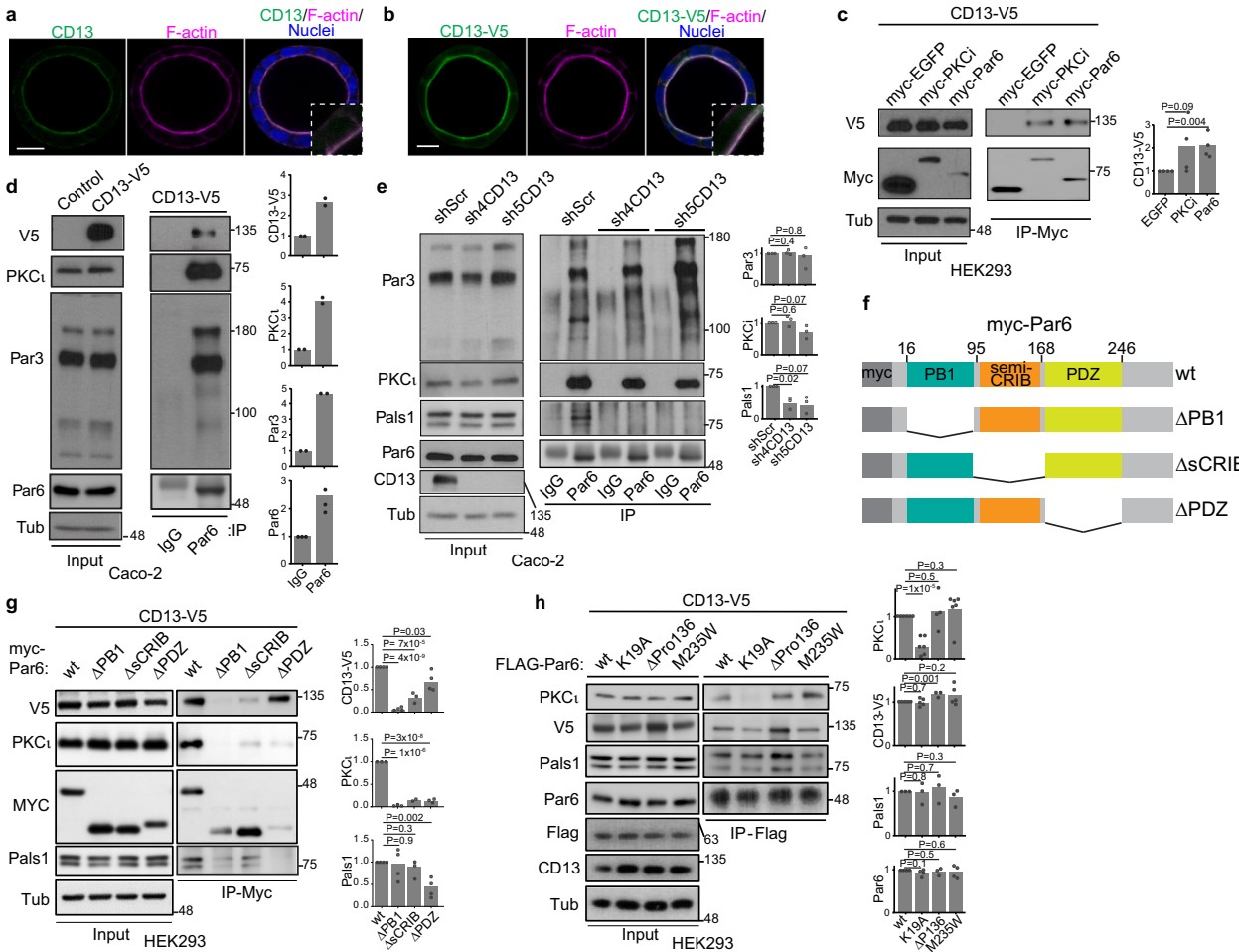

**Fig. 1 CD13 localizes at the apical domain of 3D Caco-2 cysts and associates with PB1 domain of Par6. a, b** Confocal images of polarized Caco2 cysts following 10 days in culture and immunostained for endogenous CD13 (**a**, green) or V5-tagged CD13 (**b**, green) and F-actin (phalloidin, magenta). Images are representative from three independent experiments. **c** Co-immunoprecipitation of CD13 and EGFP, PKCι, or Par6 was performed with anti-myc in HEK293 cells. The presence of CD13 in immunoprecipitates was determined by western blot analysis using anti-V5. Graphs to the right show quantification of band intensities from $n = 4$ independent experiments. **d** Immunoprecipitation was performed with IgG or anti-Par6 in stably expressed CD13-V5 Caco-2 cells and blotted for V5, PKCι, Par3, and Par6. Graphs to the right show quantification of band intensities from $n = 2$ independent experiments. **e** Immunoprecipitation was performed with anti-Par6 antibody or IgG control from shScr, sh4-CD13 and sh5-CD13 knock-down Caco-2 cell lysates and blotted for Par3, PKCι, Pals1, Par6, and CD13. Graphs to the right show quantification of band intensities from n=3 independent experiments. **f** Schematic diagram showing Par6 domains and deletion mutants. **g** Co-immunoprecipitation of CD13 and wild-type (wt) or Par6 deletion mutants was performed with anti-myc in HEK293 cells. Graphs to the right show quantification of band intensities from $n = 4$ independent experiments. **h** Co-immunoprecipitation of CD13 and wild-type (wt), or Flag-tagged Par6 containing point mutations (K19A, ΔP136, M235W) performed with anti-Flag antibodies in HEK293 cells. Graphs to the right show quantification of band intensities from independent experiments (PKCι, $n = 6$; CD13-V5, $n = 5$; Pals1, $n = 3$; Par6, $n = 4$). $p$-values were calculated by ANOVA with Tukey's posthoc test for multiple comparisons (**c, e, g, h**). Bars: **a** and **b**, 10 μm.

As an adaptor, Par6 associates with known partners through conserved protein-protein interaction domains, including PB1, semi-CRIB, and PDZ[2]. To determine if the association of CD13 with Par6 was dependent on these regions, we expressed Par6 variants with internal deletions of each domain (Fig. 1f) and tested their ability to associate with CD13. As expected, Par6$^{\Delta PB1}$ and Par6$^{\Delta PDZ}$ disrupted interactions with aPKC and Pals1, respectively (Fig. 1g). All three deletion mutants reduced the association of Par6 with CD13, however, whereas CD13 retained some capacity to associate with Par6$^{\Delta sCRIB}$ and Par6$^{\Delta PDZ}$, it was not detected in association with Par6$^{\Delta PB1}$ (Fig. 1g).

PB1 domains form heterodimeric (PB1-PB1) connections between proteins and mutation of lysine 19 (Par6$^{K19A}$) in the PB1 domain of Par6 disrupts its association with aPKC[41]. Since CD13 does not have a PB1 domain, it cannot form a canonical PB1-PB1 interaction with Par6. We therefore wondered if CD13 associated with Par6 through aPKC or by a different mechanism.

To test this, we expressed and immunoprecipitated wild-type Par6 or Par6$^{K19A}$. As expected Par6$^{K19A}$ did not associate with aPKC, however, it retained an association with CD13 (Fig. 1h). Since whole domain deletions may affect the overall structure of Par6, we examined the association of CD13 with inactivating point mutations in the semi-CRIB (Par6$^{\Delta P136}$) and PDZ (Par6$^{M235W}$) domains[41], but they did not significantly impact the association of CD13 with Par6 (Fig. 1h). Therefore, CD13 associates with Par6 through a PB1 domain-dependent mechanism that does not require aPKC binding to Par6. Together, these data demonstrate that CD13 localizes to the apical membrane and is part of an apical complex with Par6.

**CD13 is required to orient apical-basal polarity in Caco-2 cells.** A role for CD13 in cell polarity is not known. To investigate if CD13 is involved in luminogenesis, we examined cyst formation of Caco-2 cells expressing control shRNA (shScr) or two-

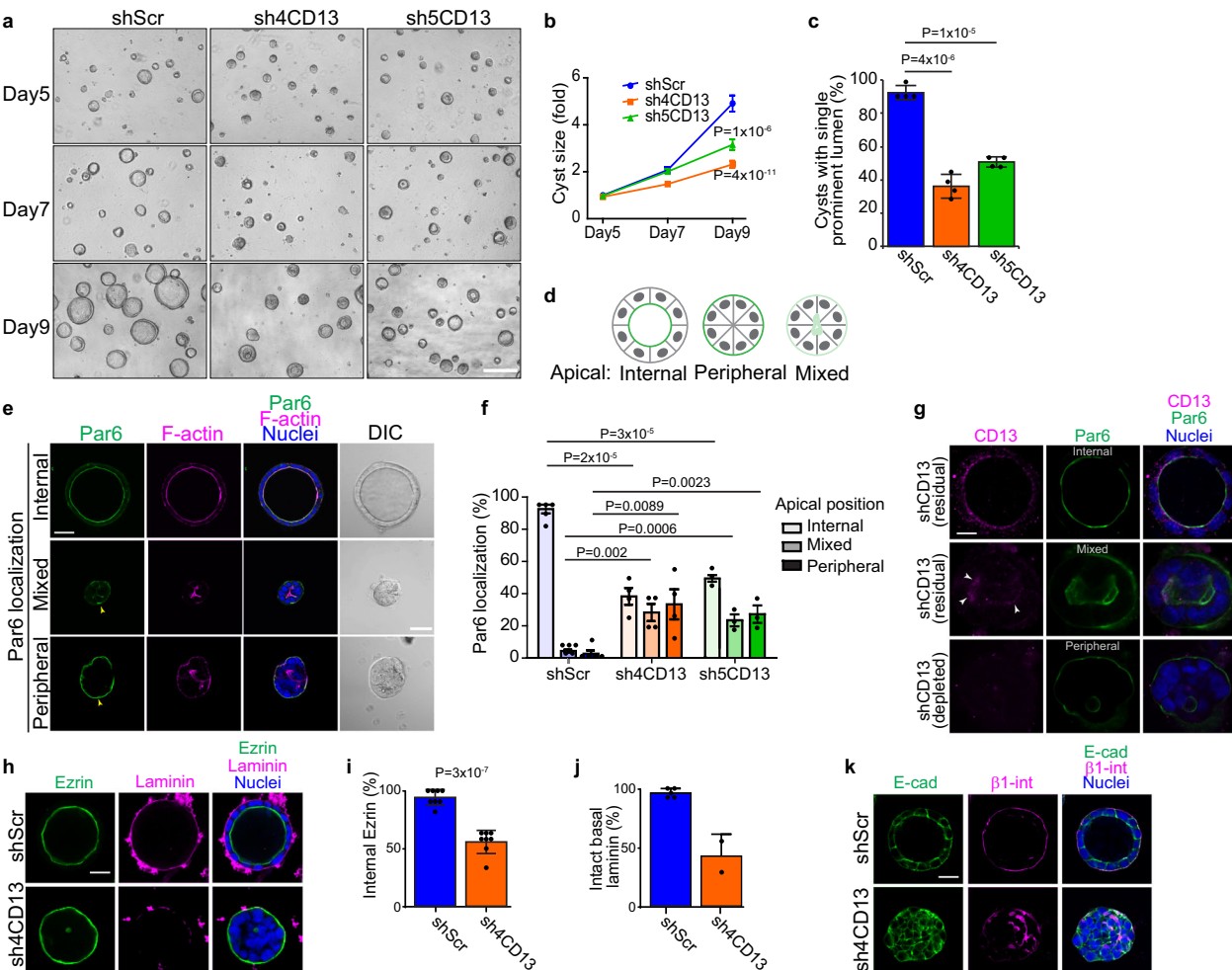

**Fig. 2 CD13 knockdown disrupts the orientation of apical-basal polarity. a** Phase contrast images showing the phenotype of shScr, sh4- and sh5-CD13 knock-down in 3D Caco-2 cysts after 5, 7, and 9 days in culture. **b** Quantification of the size of 3D Caco-2 structures for the indicated conditions after 5, 7, and 9 days in culture. Data are reported as mean ± SEM for the indicated conditions ($n = 129$ shScr, $n = 126$ sh4-CD13, and $n = 160$ sh5-CD13 cysts) from a single representative experiment. Comparisons of multiple means were performed by ANOVA using Tukey's post-hoc test on data from day 9. **c** Quantification of cysts with single prominent lumen of shScr ($n = 806$), sh4-CD13 ($n = 961$) and sh5-CD13 ($n = 1312$) knock-down in 3D Caco-2 cysts on day 10. Individual dots represent mean values from each of 4 independent experiments. **d** The schematic diagram showing different phenotypes (internal, peripheral, mixed) of apical Par6 observed. **e** Confocal images of Par6 (green) and F-actin (magenta) showing different Par6 localization (internal, peripheral, mixed) in 3D Caco-2 cysts. Yellow arrow heads indicate peripheral Par6 localization. **f** Quantification of Par6 localization (internal, peripheral, mixed) in Caco-2 structures for the indicated conditions ($n = 358$ shScr, $n = 336$ sh4-CD13, and $n = 187$ sh5-CD13 cysts) after 10 days in culture. Dots represent the mean from each of 4 independent experiments. **g** Confocal images of shCD13-expressing Caco-2 cells immunostained for endogenous CD13 (magenta) and Par6 (green) to visualize residual/depleted CD13 in individual cysts with different phenotypes. **h** Confocal images CD13 knock-down Caco-2 cysts immunostained for Ezrin (green) and laminin (magenta) showing cells with inverted apical-basal polarity. **i** Quantification of internal Ezrin in shScr ($n = 168$) and sh4CD13 ($n = 150$) 3D Caco-2 cysts after 10 days in culture. Dots represent means from each of 8 independent experiments. **j** Quantification of intact basal laminin in shScr ($n = 71$) and sh4CD13 ($n = 35$) Caco-2 cysts after 10 days in culture. Dots represent means from each of two independent experiments. **k** Images for E-cad (green) and β1-integrin (magenta) showing cell polarity are disrupted in CD13 knock-down 3D Caco-2 cysts. Data are represented as mean ± SEM (**b**, **f**) or mean ± SD (**c**, **i**, **j**). p-values were calculated by ANOVA with Tukey's posthoc test for multiple comparisons (**b**, **c**, **f**) or unpaired two-tailed student's t-test (**i**). Images are representative of two (**k**) three (**a**), or four (**e**, **g**, **h**) independent experiments. Bars: **a**, 200 μm; **e**, 50 μm; **g**, 20 μm; **h** and **k**, 30 μm.

independent shRNA (sh4CD13 and sh5CD13), which reduce CD13 protein expression (Fig. 1e). After 9 days in culture, control spheroids formed cysts with a single prominent lumen, whereas CD13-depleted spheroids were significantly smaller and fewer were able to form a single prominent lumen (94% for shScr; 38-43% for shCD13; Fig. 2a–c). CD13-knockdown spheroids had a small reduction in viability (20% of spheroids had a single apoptotic cell) associated with a minor, but statistically insignificant reduction in the number of cells per spheroid and reduced (Supplementary Fig. 2a, b) indicating that the smaller size is predominantly due to loss of the luminal cavity. To investigate

the effect on apical-basal polarity, we immunostained cysts for Par6. Whereas control cysts predominantly localized Par6 internal at the central lumen, CD13-depleted cysts showed diverse localization patterns for Par6 (Fig. 2d). Internal (luminal) apical staining was observed in some CD13-depleted structures, which had a single prominent lumen, similar to control cysts (Fig. 2e, f). A second phenotype exhibited Par6 localization to the outer edge of CD13-depleted structures (peripheral). Finally, some CD13-depleted structures displayed an intermediate of these two phenotypes (mixed) (Fig. 2d–f). In mixed cases, we observed multiple microlumen or misshapen collapsed lumen with weak staining of

apical markers (Fig. 2e). The range of phenotypes with varying degrees disruption to polarity prompted us to examine whether there was residual CD13 in Caco-2 structures with less severe phenotypes. Indeed, we observed residual endogenous CD13 at the lumen of cells with a prominent lumen, and weak residual CD13 at the small apical region in mixed cysts (Fig. 2g). Therefore, we conclude that the severity of phenotypes in individual Caco-2 cysts is dependent on the level of CD13 expression. We further confirmed apical orientation to the periphery of CD13-depleted Caco-2 cysts with an independent apical marker, Ezrin (Fig. 2h, i).

In normal polarized epithelial cells, apical and basal markers are mutually exclusive. Since we frequently observed apical markers mis-localized to the outer edge of CD13-depleted cell structures, we wondered if there was mixing of apical and basal proteins or whether apical and basal membranes remained mutually exclusive. β1-integrin-containing complexes localizes to the basal surface and associate with ECM proteins like laminin[9], which was observed in control Caco-2 cysts. In contrast, we observed that β1-integrin was often mis-localized to intercellular patches and laminin was not sequestered around the periphery of CD13-depleted cysts (Fig. 2h, k). E-cadherin was localized to cell-cell contacts in both control and CD13-depleted cells and therefore not affected by CD13-depletion. Interestingly, in some cases we observed both internal and peripheral apical localization in different cells within the same 3D structure, indicating that polarity orientation is not coordinated between adjacent cells (Supplementary Fig. 2c). Collectively, these results indicate that CD13 is required to establish the correct orientation of apical-basal polarity and subsequent lumen formation in Caco-2 cysts.

**CD13 is required to position an apical membrane-initiation site between cells.** Lumen formation occurs through initial specification of an apical membrane site that matures into a pre-luminal apical patch (PAP), and then expands to create a luminal cavity[16]. To determine if CD13 was involved in early stages of apical membrane establishment, we first examined CD13 localization in pre-luminal stages when apical-membrane specification occurs. Prior to apical membrane initiation, we observed V5-tagged CD13 at the cell-cell adhesion that was decorated with E-cadherin (Supplementary Fig. 3a). The apical membrane initiation site (AMIS) is established by recruitment of apical membrane determinants with coincident displacement of E-cadherin. At this stage, we observed recruitment of CD13 at focal sites of E-cadherin displacement (Supplementary Fig. 3aii, white arrow). The apical foci of CD13 expanded to form a pre-luminal apical patch (PAP), which was strongly anti-correlated with E-cad localization (Supplementary Figs. 3a, b).

Since CD13 can associate with Par6, we also investigated their co-localization during apical membrane initiation. Par6 did not colocalize with CD13 at the cell-cell adhesion but they did colocalize at the periphery and at the apical site during stage AMIS and PAP stages (Fig. 3a). Similar results were obtained with Pals1 (Supplementary Fig. 3c). We confirmed the localization of endogenous CD13 to adhesion, AMIS, PAP, and the apical surface of the emerging lumen, thus validating the localization patterns observed with CD13 tagged on the extracellular C-terminus (Fig. 3b). The unique localization of CD13 at the cell adhesion followed by its accumulation at the AMIS prompted us to investigate the dynamics of this event in more detail using time-lapse imaging. For this, we expressed mCherry-tagged CD13 (CD13-mCh) and GFP-tagged Par6 (GFP-Par6) in Caco-2 cells, then imaged from single cell stage (before apical membrane initiation) through lumen formation. After the first cell division, CD13-mCh transiently localized to the cell-cell contact surface

(Fig. 3c), consistent with our data from fixed images (Fig. 3a, b). Cell contact-associated CD13-mCh was then dispersed diffusely in the cytoplasm as small puncta, which coalesced to eventually form a compacted blob (AMIS) that subsequently expanded to form a nascent lumen (Fig. 3c). The dispersed CD13-mCh puncta were mostly GFP-Par6-negative, and coalescence of these puncta into a larger single patch was co-incident with GFP-Par6 accumulation (Fig. 3d; Supplementary Video 1). This result suggests that Par6 may help stabilize CD13 during apical membrane specification and lumen formation. Therefore, these data demonstrate that CD13 accumulates at the site for apical membrane initiation before apical proteins.

To explore a role for CD13 in apical membrane initiation we examined the consequences of CD13-depletion on cells 24 hrs after seeding. At this time, cells have divided (i.e., 2-cell stage) and 80% of control cells (shScr) have initiated apical membrane specification, as marked by Par6 accumulation at the interior (Fig. 4a, b). At this time, some control cells had minor residual Par6 on the periphery, which represent 3D structures that had not yet fully polarized (Figs. 4a, c and 3a). In contrast, 40-60% of CD13-depleted cells failed to accumulate any Par6 at an internal site and >90% instead retained Par6 on the periphery (Fig. 4a–c). This does not result from delayed apical specification at internal sites in 2-cell structures, because we also observed Par6 on the periphery in 4-cell structures when a lumen has initiated in controls, and after 10-days in culture, when control structures have a well-established lumen (Figs. 2e, f, 4a). Similar effects were observed with independent apical markers, Pals1 and Ezrin, and tight junction maker ZO1 (Supplementary Fig. 4a–f).

In the subset of CD13-depleted 2-cell structures with internal Par6 accumulation, we also observed robust Par6 staining on the periphery (Fig. 4d). Moreover, we observed that the position of internal Par6 accumulation at internal sites was offset to the side of the 2-cell structure and not between the nuclei (Fig. 4d). To quantify this, we calculated the Par6 Offset Index, in which a value of 1 represents the middle of the structure, and values approaching 0 are displaced towards the periphery (Fig. 4e). This reveals that whereas control cells typically accumulate Par6 at the center of 2-cell structures (Offset Index$_{shScr}$ = 0.91 ± 0.1), internal Par6 accumulation is offset towards the periphery of CD13-depleted 3D structures (Offset Index$_{shCD13-4}$ = 0.19 ± 0.16); Fig. 4e, f).

Previous studies reported that the cytokinetic midbody positions the apical membrane initiation site[18,42,43]. Therefore, we examined whether the midbody position was affected by CD13 knockdown. We did not observe apoptosis or multi-nucleated cells in control or CD13-knockdown, indicating that cell-division per se was not blocked under these conditions. In control cells, the midbody formed between nuclei at the middle of 2-cell structures (Offset Index$_{shScr}$ = 0.82 ± 0.19) with Par6 accumulating at the center of the midbody. In contrast the midbody in CD13-depleted cells was displaced towards the cell edge (Offset Index$_{shCD13}$ = 0.35-0.45 ± 0.32-0.35) and Par6 did not accumulate at the center of the midbody (Fig. 4g–i) indicating that trafficking of proteins to initiate apical membrane at the midbody was impaired. Collectively, these data indicate that CD13 is required for accumulation of apical membrane components to the center of the midbody.

**CD13 is required for trafficking apical proteins through Rab11-vesicles.** The Rab family of GTPases have important roles in vesicle trafficking, and the Rab11 sub-family plays a vital role in apical endosome recycling and lumenogenesis[12–17]. In migrating cells, CD13 was also shown to regulate integrin recycling by Rab11[33]. To determine if CD13 may influence Rab11-

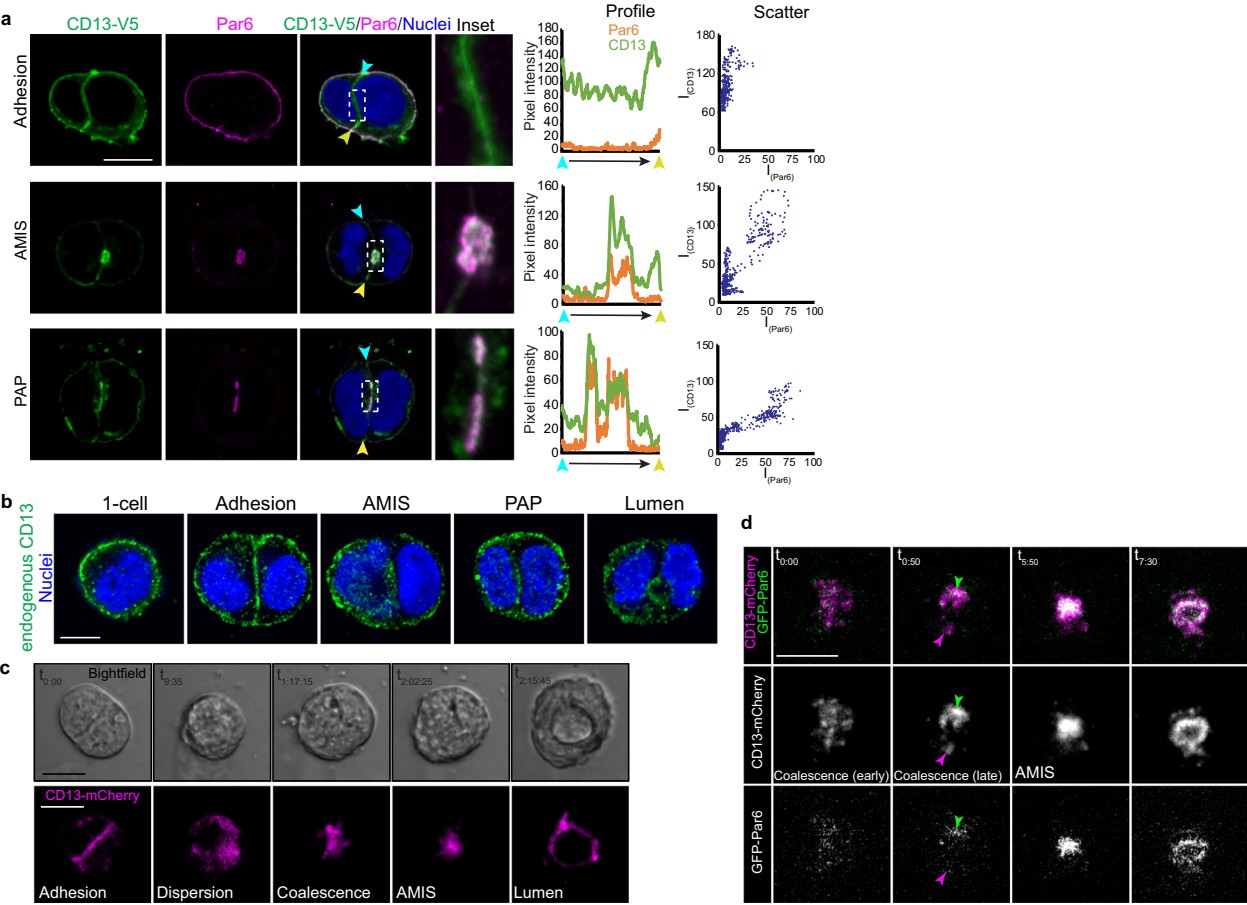

**Fig. 3 CD13 is recruited to the apical membrane-initiation site before Par6 in 3D Caco-2 cysts. a** Confocal images of 2-cell Caco-2 spheroids immunostained for CD13-V5 (green) and Par6 (magenta) showing the localization of CD13 and Par6 at adhesion, AMIS, and PAP stages. Middle - profiles depicting fluorescent intensity (8-bit) of CD13 and Par6 along the cell-cell edge from blue to yellow arrows. Right - Scatter plots showing the relationships between Par6 and CD13 pixel intensities. **b** Confocal images from Caco-2 cells immunostained for endogenous CD13 during polarization and early lumen formation stages (1-cell, adhesion, AMIS, PAP, lumen). **c**, **d** Caco-2 cells were lentivirus-infected to express CD13-mCherry and EGFP-Par6 vectors in 2D culture, then transferred to 3D culture to initiate cell polarization. Time-lapse confocal images were captured every 25 min. Between adhesion and AMIS stages, CD13 disperses into puncta that coalesce into an AMIS that subsequently opens to form a lumen (**c**). During coalescence, CD13-puncta accumulation precedes incorporation of Par6 at the AMIS (**d**). Images are representative from three independent experiments. Bars: **a**–**d**, 10 μm.

dependent vesicle trafficking during epithelial polarization, we first investigated the localization of CD13 and Rab11 during apical membrane initiation in Caco-2 cells. In single cells prior to polarization, we observed CD13 and Rab11-positive vesicles near the plasma membrane (Fig. 5a). Analysis of local peak intensities indicate that CD13 is in close proximity, but does not fully superimpose with Rab11-positive vesicles (Supplementary Fig. 5a). During apical membrane initiation (AMIS), Rab11 was observed diffusely surrounding CD13, which became more concentrated to a ring of Rab11 that decorated the edge of CD13 in the pre-luminal apical patch (PAP) (Fig. 5a, Supplementary Fig. 5a). Moreover, CD13 puncta coalesce during AMIS formation initially without local Rab11 enrichment (Fig. 5b), indicating that CD13 accumulation at the future apical membrane site precedes Rab11. Therefore, we examined whether Rab11 trafficking was influenced by CD13 during apical membrane initiation and lumen formation. In control cells, Rab11 was distributed near the plasma membrane in single cells, which was redistributed to areas between the nuclei and accumulated adjacent to Par6-positive apical membrane in 2-cell structures and at later stages (>2 cells) when the lumen begins to form (Fig. 5c, d). In CD13-depleted cells, Rab11 was also distributed near the plasma membrane in single cells but was not positioned between nuclei

of 2-cell and >2-cell structures, instead it accumulated with no obvious pattern, and often at multiple sites (Fig. 5c, d). Using live imaging we observed that accumulation of Rab11 occurred within a similar time-scale between control and CD13-depleted samples (Supplementary Fig. 5b; Supplementary Videos 2 and 3), suggesting that CD13 does not control the kinetics of Rab11 accumulation. Moreover, Rab11-vesicles in CD13-deficient cells were initially directed towards the center, but were not retained and redistributed towards the periphery (Supplementary Fig. 5b). This indicates that CD13 is necessary for correct trafficking of Rab11 recycling endosomes.

Interestingly, we observed that whereas Par6 colocalized with a subset of Rab11 vesicles and was redistributed to the center of control cells, Par6 was retained on the peripheral plasma membrane and was not internalized in CD13-deficient cells (Fig. 5c). Previous reports indicate that apical proteins that fail to accumulate at the AMIS are recycled to the plasma membrane[44]. Therefore, we examined Par6 localization in cysts with Rab11 accumulation at the presumptive AMIS to determine if it was initially internalized and returned, or remained at the periphery. Strikingly, Par6 was retained at the cell periphery and excluded from the cytoplasm in CD13-depleted cells (Fig. 5e–g). This indicates that Rab11 recycling endosomes are devoid of at least

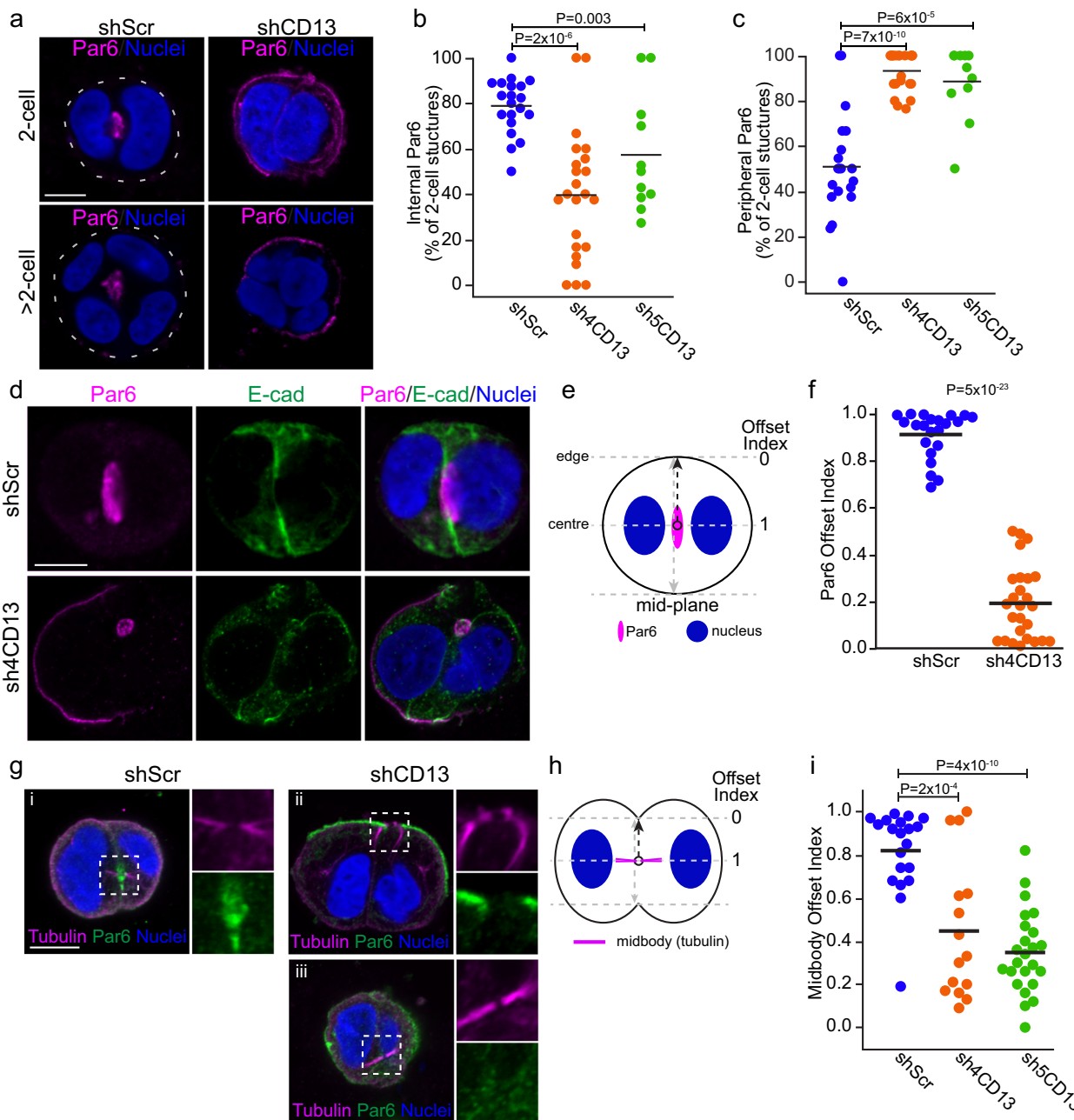

**Fig. 4 Depletion of CD13 disrupts midbody position and apical protein accumulation to the midbody. a** Confocal images of Par6 (magenta) in control (shScr) and CD13 knock-down 3D Caco-2 structures. **b**, **c** Quantification of the percentage of 3D structures with internal (**b**) and peripheral (**c**) Par6 localization in control shScr ($n = 20$), sh4CD13 knock-down ($n = 23$) and sh5CD13 knock-down ($n = 13$) different fields of view. Data are representative of two independent experiments. **d** Representative confocal images of Par6 (magenta) and E-cad (green) in control and CD13 knock-down 3D Caco-2 structures. **e** Diagram showing the offset index of Par6 in 2-cell 3D Caco-2 structures. A value of one indicates midbody position at the center of the 3D structure, and values approaching 0 represent midbody positioned closer to the edge. **f** Quantification of Par6 offset index from shScr ($n = 22$) and sh4-CD13 knock-down ($n = 26$) 2-cell 3D Caco-2 structures. Data are representative of three independent experiments. **g** Confocal images of tubulin (magenta) and Par6 (green) in 2-cell 3D Caco-2 structures. **h** Diagram showing the offset index of the midbody for the edge and the center (where midbody locates) on 2-cell structure. **i** Quantification of midbody offset index from shScr ($n = 20$), sh4CD13 knock-down ($n = 15$) and sh5CD13 knock-down ($n = 24$) on 2-cell 3D Caco-2 structures. Data are representative of three independent experiments. *p*-values were calculated by ANOVA with Tukey's posthoc test for multiple comparisons (**b**, **c**, **i**) or unpaired two-tailed student's t-test (**f**). Images are representative from two (**a**) or three (**d**, **g**) independent experiments. Bars: **a**, **d**, **g**, 10 μm.

some apical proteins, which fail to internalize from the peripheral plasma membrane.

To verify that Rab11 is required to redistribute Par6 from the peripheral plasma membrane to the internal AMIS, we knocked down Rab11 from Caco-2 cells and examined Par6 localization at the 2-cell stage. Whereas Par6 was efficiently translocated to

internal apical membrane patch in control cells, Par6 remained on the peripheral plasma membrane in Rab11-knockdown cells (Fig. 5h). Since both Rab11 and CD13 are required to redistribute Par6 and other apical proteins to establish an apical membrane site and they are in proximity near the cell periphery, we asked whether they could be physically connected. To test this, we first

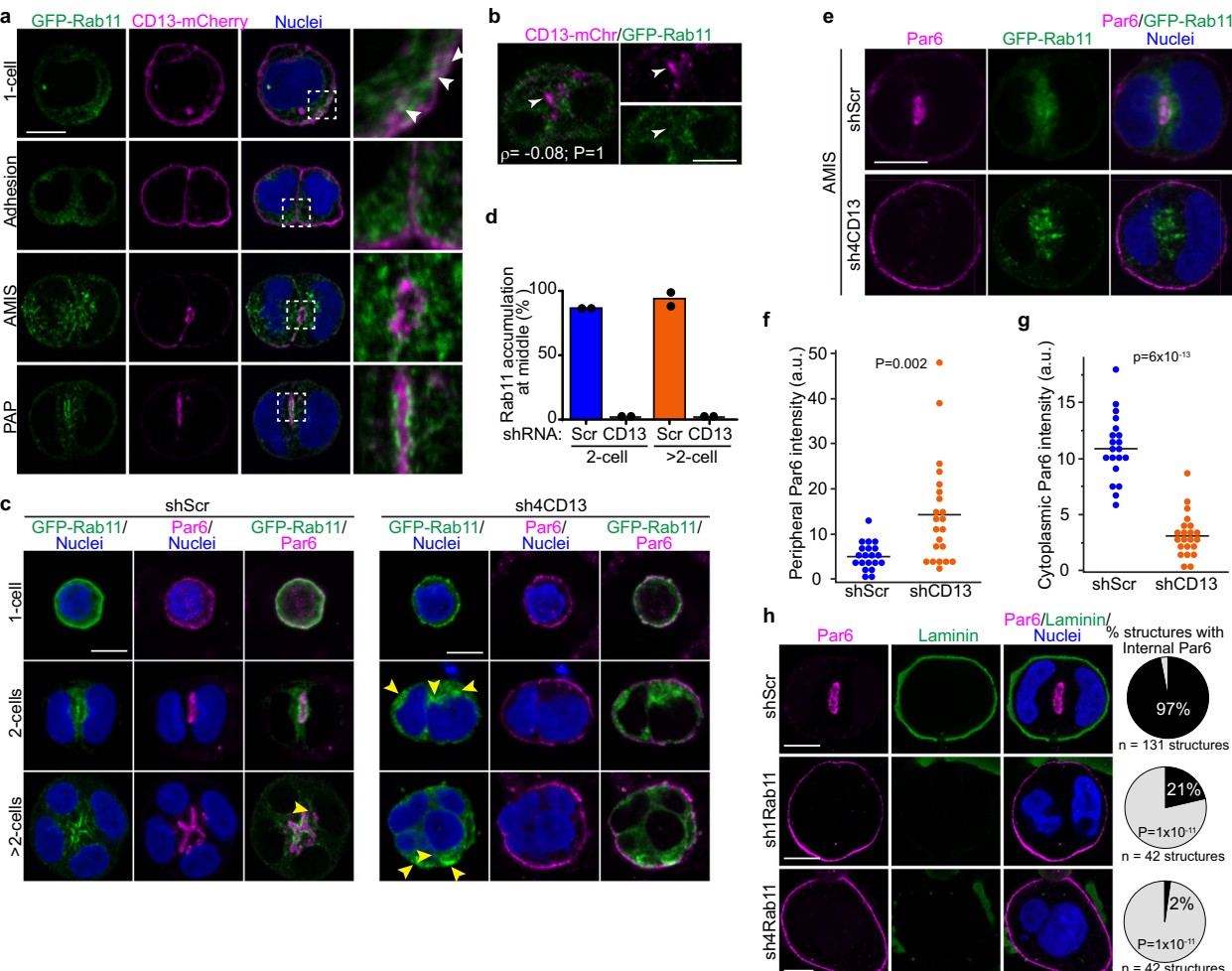

**Fig. 5 CD13 Knockdown disrupts Rab11-dependent early polarization in Caco-2 cells. a** Confocal images showing the localization of CD13-mCherry (magenta) and GFP-Rab11 (green) at 1-cell, and adhesion, AMIS, and PAP stage in 2-cell 3D Caco-2 structures. White arrows show the proximity of CD13 puncta with Rab11-positive vesicles near the plasma membrane. **b** Confocal images of Caco-2 cells showing CD13-mCh (magenta) and GFP-Rab11 (green) vesicles during AMIS formation. Object-based colocalization was performed for CD13 and Rab11, and produced a correlation efficient ρ = -0.08. Arrows show local sites of CD13 accumulation. **c** Confocal images of GFP-Rab11 (green) and Par6 (magenta) showing their localization in shScr and shCD13 Caco-2 cells. Yellow arrows show regions of concentrated Rab11-positive vesicles. **d** Quantification of the percentage of Rab11 accumulation at the middle of shScr ($n = 92$) and shCD13 ($n = 24$) at 2 cell and >2 cell 3D Caco-2 structures. Dots represent means from each of two independent experiments. **e** Confocal images of 3D Caco-2 structures at the AMIS stage showing localization of GFP-Rab11 (green) and Par6 (magenta). **f, g** Quantification of Par6 localization to the periphery (**f**) or cytoplasm (**g**) in shScr ($n = 20$) and shCD13 ($n = 22$) Caco-2 cells at the AMIS stage. Dots represent individual Caco-2 structures from a single experiment. **h** Confocal images of control (shScr) and Rab-down (sh1-Rab11, sh4-Rab11) 3D Caco-2 cells immunostained for Par6 (magenta) and laminin (green). Right – quantification of structures with internal Par6 staining as a measure of normal or inverted polarity. The indicated number of structures were measured from ($n = 11$ shScr, $n = 11$ sh1Rab11, and $n = 9$ sh4Rab11) experiments. p-values were determined by ANOVA with Tukey's posthoc test for multiple comparisons (**d**, **h**) or unpaired two-tailed student's t-test (**f**, **g**). Images are representative from two (**c**) or three (**a**,**b**, **e**, **h**) independent experiments. Bars: 10 μm.

co-expressed GFP-Rab11 and CD13-V5 in HEK293 or Caco-2 and performed co-immunoprecipitation experiments. From this data, we observed that CD13 was able to pull-down Rab11 supporting a potential physical connection between CD13 and Rab11-containing structures (Supplementary Figs. 5c, 6c). Since immunoprecipitation experiments in cell lysates can lack spatial regulation and potentially overestimate associations, we performed temporal image correlation microscopy (tICM) to investigate the correlated dynamics of GFP-tagged Rab11 and mCherry-tagged CD13 in time and space near the plasma membrane using total internal reflectance fluorescence (TIRF) microscopy. ICM relies on the coupled fluctuations of two different-colored fluorescent molecules moving in and out of the illumination beam and can determine if two different molecules associate physically, but do not indicate direct binding. Whereas

the cross-correlation amplitude between GFP control and CD13-mCherry was negligible, we observed a high cross correlation amplitude between GFP-Rab11 and CD13-mCherry (Supplementary Figs. 5d–f), indicating that CD13 and Rab11 are physically coupled near the plasma membrane. These results indicate that CD13 and Rab11 are required to redistribute apical proteins from the peripheral plasma membrane to establish an internal apical membrane site.

**CD13 acts upstream of Rab35 in apical trafficking to the AMIS.** Rab35 was shown to act as a receptor to capture vesicles containing apical proteins arriving at the emerging AMIS; in the absence of Rab35 vesicles recycle to the plasma membrane resulting in inverted polarity[44]. Given the similarities between CD13- and Rab35-knockdown phenotypes and that CD13 puncta

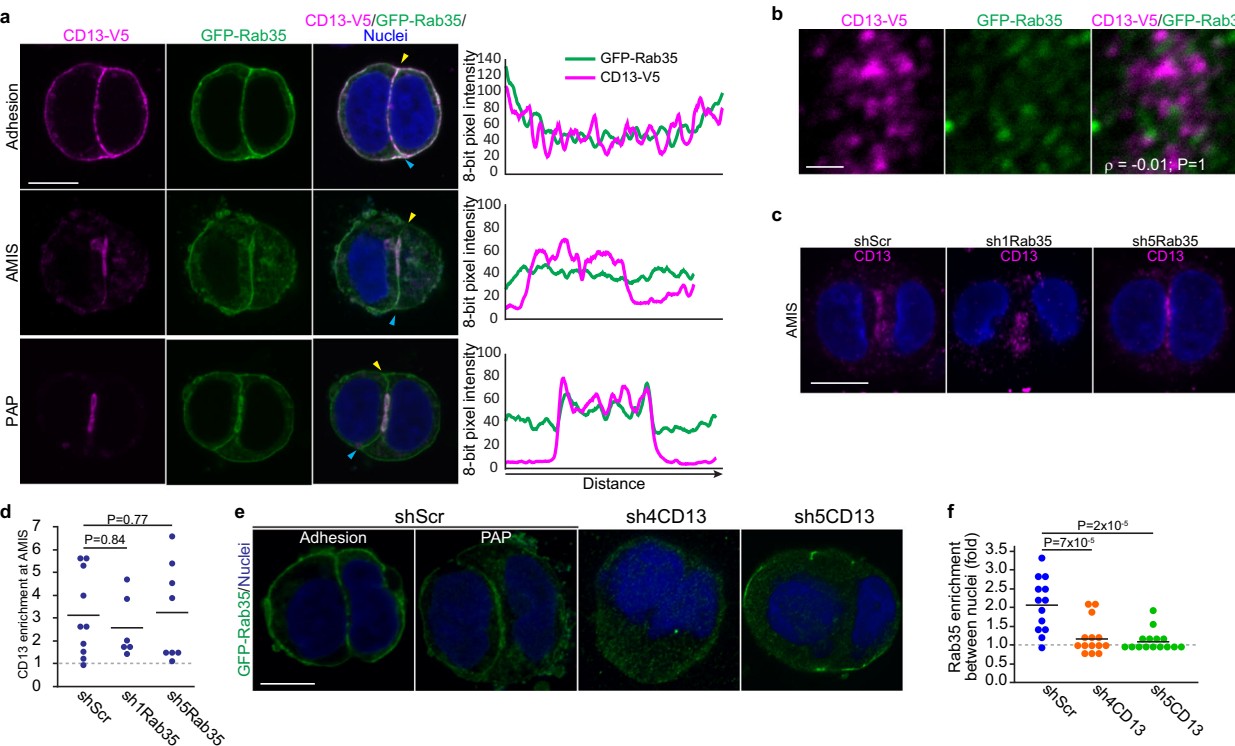

**Fig. 6 CD13 acts upstream of Rab35 in apical specification. a** Confocal images of Caco-2 cells grown in 3D culture for 1 days then immunostained for V5-tagged CD13 (magenta) and GFP-Rab35 (green) at initiation stages of polarization (Adhesion, AMIS, PAP). Line tracings showing CD13 and Rab35 pixel intensity along the cell-cell contact surface from yellow to cyan triangles are shown to the right. **b** Confocal images of Caco-2 cells showing CD13-V5 (magenta) and GFP-Rab35 (green) vesicles. Object-based colocalization was performed for CD13 and Rab35 and produced a correlation efficient $\rho = -0.01$. **c** Confocal images of control (shScr) or Rab35 knock-down (sh1-Rab35, sh5-Rab35) Caco-2 cells immunostained for endogenous CD13 (magenta). **d** Quantification of internuclear enrichment of CD13. Dots represent individual Caco-2 structures from a single experiment. **e** Confocal images of control (shScr) and CD13 knockdown (sh4-CD13 and sh5-CD13) Caco-2 cells showing GFP-Rab35 direct fluorescence. **f** Inter-nuclear enrichment of GFP-Rab35 was calculated as the mean internuclear intensity divided by mean cytoplasmic intensity excluding the inter-nuclear region for the conditions indicated ($n = 13$ shScr, $n = 14$ sh4CD13, $n = 15$ sh5CD13 cell structures). Data are representative of two independent experiments. p-values were calculated by ANOVA with Tukey's posthoc test for multiple comparisons (**d**, **f**). Images are representative from 2 independent experiments. Bars: **a**, **c**, **e**, 10 μm; **b**, 2 μm.

coalesce at the AMIS, we examined a potential relationship between CD13 and Rab35. First, we examined the localization of CD13-V5 and GFP-Rab35 in Caco-2 cells at the 2-cell stage. Both CD13 and Rab35 are initially present at the adhesion, but CD13 is concentrated at the emerging AMIS before Rab35 (Fig. 6a). Rab35 subsequently becomes concentrated to a region co-incident with CD13. We examined CD13 and Rab35 vesicles at the time of AMIS formation and observed no overlap between the two (Fig. 6b), suggesting that CD13 accumulation at the AMIS is likely independent of Rab35. To verify this, we depleted Rab35 from Caco-2 cells and evaluated localization of endogenous CD13. Strikingly, we observed that similar to controls, CD13 retains the capacity to accumulate at internal patches in Rab35-knockdown cells (Fig. 6c, d). In most cases, the patches appear loose and uncompacted, consistent with a role for Rab35 to receive apical vesicles necessary to form a stable apical domain. To determine if alternatively, CD13 may be required for Rab35 concentration at the emerging AMIS, we examined Rab35 localization in CD13-knockdown cells. Indeed, we observed that Rab35 failed to localize properly at the plasma membrane in CD13-deficient cells, indicating that Rab35 localization is dependent on CD13 (Fig. 6e, f). Together, these data support that CD13 accumulates to internal apical initiation sites prior to Rab35 and Rab11.

**The CD13 intracellular domain is required to orient apical-basal polarity.** Next, we examined regions of CD13 that were

required for polarization in Caco-2 cells. CD13 has an extra-cellular M1 peptidase domain and a short N-terminal intracellular domain. To evaluate whether peptidase activity was necessary for CD13 polarity functions, we conducted rescue experiments in CD13 depleted cells using vectors that expressed wild-type CD13 or a series of catalytically inactive mutants (H388A, H392A, or E411A) and examined polarity orientation (Fig. 7a). In all cases polarity orientation was restored, indicating that CD13 catalytic activity is not essential for orienting apical-basal polarity (Fig. 7b, c, Supplementary Figs. 6a, b). Tyrosine (Y6) within the intracellular domain was previously reported to be necessary for some non-catalytic functions of CD13[31]. We therefore investigated if this residue was also necessary to establish the orientation of apical-basal polarity in Caco-2 cysts by re-expressing wild-type or Y6 mutant (Y6F) in control and CD13-depleted cells. Whereas wild-type CD13 was able to restore the correct apical polarity orientation, Y6F failed to restore polarity orientation (Fig. 7b, c). CD13 containing other point mutations within the intracellular domain were also able to restore polarity orientation, demonstrating specificity of the tyrosine residue (Fig. 7a–c, Supplementary Figs. 6a, b). CD13 was previously reported to be phosphorylated on Tyr6 in monocytes[31]. To determine if Tyr6 is phosphorylated in Caco-2 cells, we immunoprecipitated wild-type or the Y6F mutant CD13 and blotted with a pan-phospho-Tyr antibody. However, we did not detect a difference total phospho-Tyr signal between Y6F mutant and wild-type protein (Supplementary Fig. 6c). This suggests that

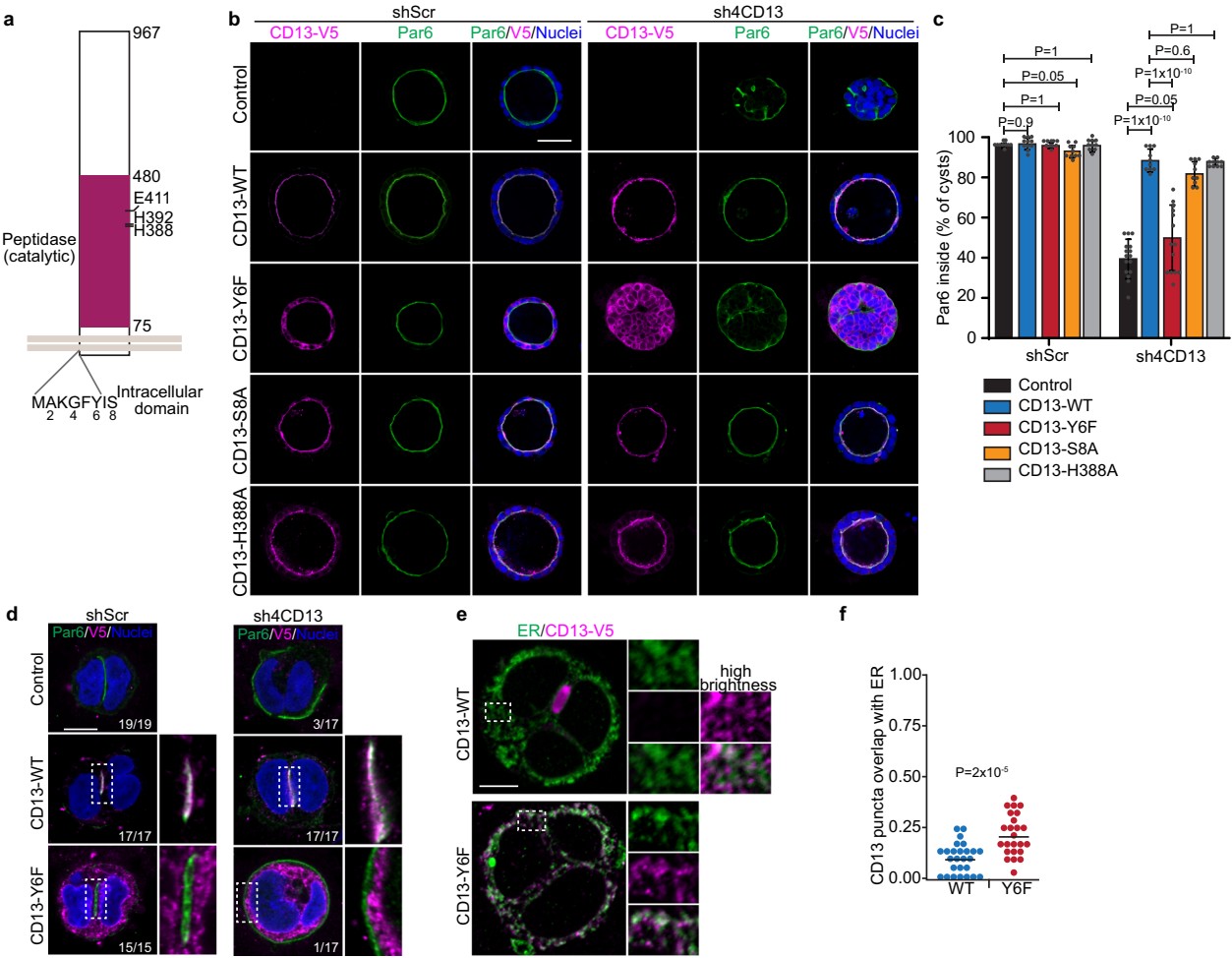

**Fig. 7 The intracellular domain of CD13 is required to maintain apical-basal polarity. a** Schematic diagram showing mutation sites at intracellular and peptidase domain of CD13. **b** Confocal images of CD13-V5 (magenta) and Par6 (green) showing the rescue phenotype in wildtype and different CD13 mutants of shScr and shCD13 Caco-2 cysts. **c** Quantification of the percentage of Caco-2 spheroids with internally localized Par6 in wildtype (shScr, $n =$ 515; shCD13, $n =$ 292), CD13-Y6F (shScr, $n =$ 462; shCD13, $n =$ 433), CD13-S8A (shScr, $n =$ 541; shCD13, $n =$ 333), and CD13-H388A (shScr, $n =$ 452; shCD13, $n =$ 336) of shScr ($n =$ 428) and shCD13 ($n =$ 492) Caco-2 cysts. Dots represent mean values ± SD from each of 10 experiments. $p$-values were calculated by ANOVA with Tukey's posthoc test for multiple comparisons. **d** Images for CD13-V5 (magenta) and Par6 (green) showing the rescue phenotype in wildtype and CD13-Y6F mutant of shScr and shCD13 Caco-2 in 2-cell structures. Numbers in the lower right indicate the fraction of cells analyzed with Par6 localized between the nuclei. **e** Caco-2 cells expressing CD13-WT or CD13-Y6F were immunostained for CD13-V5 (magenta) and Calreticulin to label the endoplasmic reticulum (ER) (green). **f** Quantification of the proportion of CD13 puncta that overlap with sites of ER-positivity. Dots represent individual Caco-2 structures from a single experiment. $p$-values were calculated with an unpaired two-tailed student's t-test. Images are representative from two (**b**, **d**) independent experiments. Bars: **b**, 50 µm; **d, e**, 10 µm.

Tyr6 is not phosphorylated in Caco-2 cells. Both wild-type and Y6F-CD13 show similar levels of phospho-Tyr, which may result from phosphorylation on one of the 44 extracellular Tyr-residues. Interestingly, secreted tyrosine kinases are known to phosphorylate secreted and endoplasmic reticulum-resident substrates[45], however future studies will be required to investigate tyrosine phosphorylation of the CD13 extracellular domain.

We extended these studies to evaluate CD13-Y6F at early stages when apical membrane is initiated at internal sites (2-cell). Consistent with our above data, the CD13-Y6F mutant was deficient in restoring internal apical sites in CD13-depleted cells (Fig. 7d). Interestingly, we did not observe mis-oriented polarity in control cysts with endogenous CD13 that express CD13-YF6, indicating that it does not behave as a dominant negative protein (Fig. 7b–d). Moreover, we observe in both early (2-cell) and mature (>10 cells) 3D structures, that CD13-Y6F was localized to intracellular puncta instead of the outer plasma membrane and did not overlap with Par6 (Fig. 7d). CD13-Y6F had a reduced

association with Par6 consistent with its failure to localize to the apical membrane (Supplementary Fig. 6c).

The staining pattern of CD13-Y6F is reminiscent of endoplasmic reticulum (ER) and may reflect CD13 becoming trapped. We therefore examined colocalization of CD13-Y6F and the ER-marker Calreticulin to determine the number of CD13 puncta that overlapped with the ER. Whereas 9% of CD13 puncta colocalized with ER in WT cells, this was increased to 20% in Y6F mutants (Fig. 7e, f). This indicates that there is modest effect on ER retention of Y6F, but the majority of CD13 is not trapped in the ER. Therefore, these results indicate that intracellular domain of CD13 is required for its localization to the plasma membrane and association with Par6.

## Discussion

Collectively, our results demonstrate that CD13 (also known as APN and ANPEP) controls epithelial lumen formation by

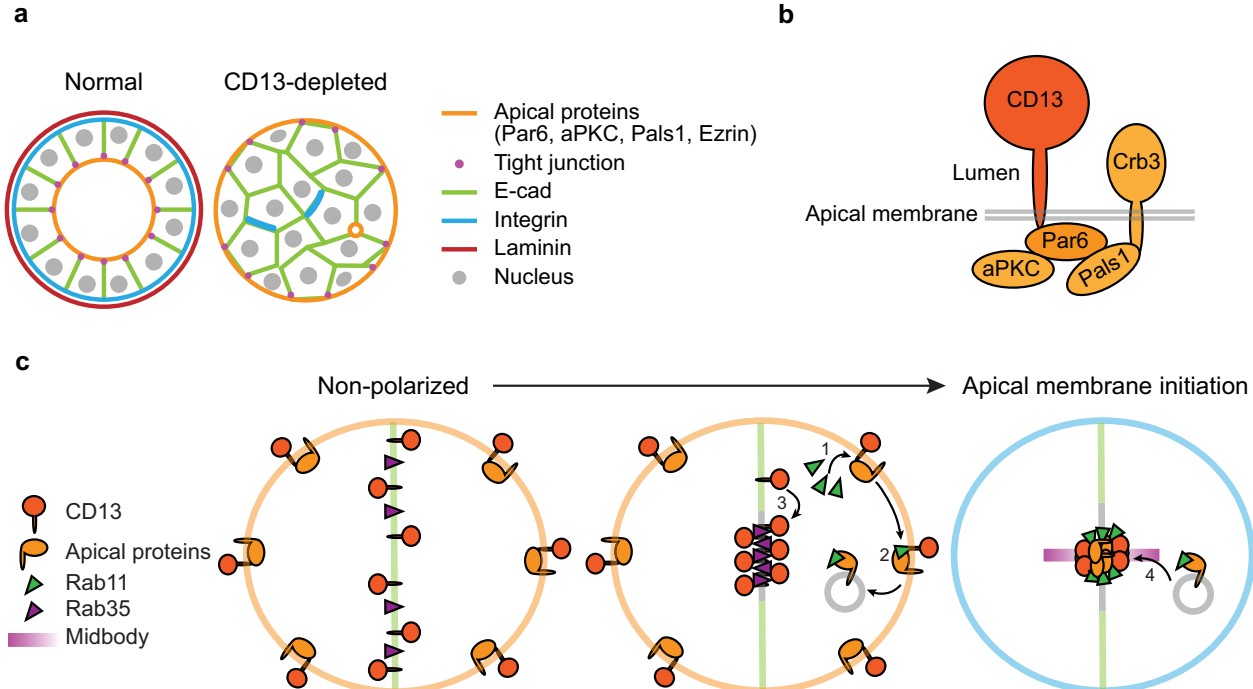

**Fig. 8 Proposed model for CD13 functions in apical specification in Caco-2 cells. a** Organization of control Caco-2 3D structures displaying apical-basal polarity and epithelial organization, and CD13-depleted cells showing inverted apical-basal polarity and impaired lumen formation. **b** CD13 associates with apical complex containing Par6, aPKC, Pals1, and Crb3. **c** Proposed events by which CD13 functions to establish the orientation of apical-basal polarity. CD13 associates with apical proteins (1) and is required for their loading to Rab11-mediated endosomes (2). In parallel, CD13 establishes Rab35 at the AMIS (3) necessary to capture vesicles containing apical determinants for apical membrane initiation (4). See discussion for a further explanation.

directing the orientation of apical-basal polarity. Silencing of CD13 induced an inverted polarity phenotype in Caco-2 cysts, which was characterized by apical complexes and tight junctions positioned at the periphery of cell aggregates, while basolateral proteins were excluded from the outer edge and interactions with the matrix were impaired (Fig. 8a). The formation of a central lumen requires precise coordination of multiple intra- and intercellular events that establish a polarity axis, position apical determinants between cells, and separate cell-cell contacts to expand a central luminal cavity[10,16,46,47]. This architecture is essential for normal tissue function and epithelial homeostasis, and is frequently disrupted or lost in diseases including cancer[3,48].

Par6 can associate with two apically directed polarity complexes: The Par complex by binding Par3 at tight-junctions, or the Crumbs complex through Pals1 and Crumbs3 at the apical membrane[38,39]. Our results show that CD13 is an apical protein, which is supported by previous studies showing that CD13 localizes to the apical brush border[49,50]. We extend this to demonstrate that CD13 associates with an apical complex containing Par6, aPKC, Pals1, and Crb3, but not Par3 (Fig. 8b). This interaction requires the PB1-domain of Par6 and the intracellular domain of CD13. Although Par6 interacts with aPKC through its PB1 domain[51], a point-mutation in the PB1-domain of Par6 that disrupted its interaction with aPKC retained the capacity to associate with CD13. Moreover, deletion or mutation of other protein-protein interaction domains within Par6 that mediate its known interactions[2] also retained the ability to associate with CD13. Therefore, we propose that Par6 either binds CD13 directly through its PB1 domain, or indirectly through an unidentified intermediate. Efforts to detect a direct interaction between purified Par6 and a peptide of the CD13 intracellular domain have been unsuccessful, and we cannot distinguish these possibilities at present.

Our imaging data reveal that CD13 is dynamically localized during apical membrane initiation and lumen formation in a pattern that is distinct from other apical polarity proteins. In non-polarized cells, apical proteins are distributed to the cell periphery and re-distributed to the midbody thorough Rab11-mediated endosome recycling to form an apical membrane initiation site that marks the position of the future lumen[17,18,22,52]. While CD13 also localized to the periphery in non-polar cells, unlike other apical proteins, it temporarily localized to the cell-cell adhesion that is also positive for E-cadherin. CD13 then redistributed through vesicles that accumulated internally, prior to the delivery of other apical proteins that converged to establish an internal apical membrane site that excludes E-cadherin. These data indicate that CD13 likely exists in multiple pools that interconvert during polarization and apical membrane specification. One CD13 pool resides at the apical plasma membrane and associates with Par6 and the Crumbs complex. This pool is also juxtaposed to Rab11-containing structures and overlaps with Rab35. A second pool localizes to vesicles that do not overlap with apical proteins, Rab11, or Rab35. A third pool appears transiently on the cell-cell contact that is positive for E-cadherin and Rab35. We observe interconversion of these pools during initiation of the AMIS in which the adhesion pool of CD13 disperses into a vesicular pool that subsequently coalesce into the apical pool during AMIS formation. The mechanism that directs the interconversion between these pools is unknown, but is likely promoted by cues from cytokinesis that initiate apical membrane formation[10,44].

The transition of CD13 from the adhesion to apical site is reminiscent of basolateral-to-apical transcytosis described for a few proteins[53,54]. CFTR was shown to undergo apical-transcytosis through a Rab11-independent mechanism involving EEA1+ vesicles, raising the possibility that CD13 may redistribute to the AMIS through a transcytosis-like pathway. We show that

mutation of Tyr6 in the intracellular domain of CD13 (CD13-Y6F) is locked in a vesicular compartment that does not translocate to the plasma membrane, indicating a crucial role for the intracellular domain in controlling the distribution of CD13 to different pools in Caco-2 cells.

Rab11 is required for mitotic spindle orientation through affecting dynein-dependent endosome localization at poles[55]. Rab11 also plays an important role in regulating the trafficking of proteins for cell abscission and loss of Rab11 causes cytokinetic failure and multinucleated cells[56]. We did not observe cytokinetic deficiencies or multinucleated cells in our experiments, indicating that CD13 contributes to a subset of Rab11-dependent events, likely restricted to apical membrane trafficking in epithelial cells. However, we cannot exclude other trafficking roles for CD13 that are context specific. For example, in mouse embryonic fibroblasts, CD13 is required for endocytosis of β1-integrin to Rab11-positive endosomes during cell migration[33].

Although cell division was able to proceed in CD13-deficient cells, we did observe asymmetric placement of the midbody away from the center of cell spheroids. Interestingly, the relationship between the apical domain and midbody is bi-directional. In addition to the midbody serving to direct trafficking of apical proteins, apical proteins also direct the position of the midbody, and depletion of some apical proteins can cause displacement of the midbody that leads to ectopic lumen formation[22,57]. In the context of CD13, we speculate that apical membrane retained at the periphery of cell aggregates may cause the midbody to be pulled away from the center, although alternatives where CD13 more directly controls midbody position are also possible.

Cell division represents the earliest known symmetry breaking event to position a central lumen in 3D cysts and in vivo[56,58,59]. However, the cues that initiate internalization of apical proteins from the periphery to establish an internal apical site are currently unknown. We propose that CD13 represents this factor, which regulates the initial release of apical proteins from the periphery for loading into vesicles destined to initiate an internal apical membrane site necessary for lumen formation. In support of this model, we demonstrate that Par6 fails to redistribute to the AMIS, and instead is retained at the cell periphery in both CD13- and Rab11-depleted cells. CD13 was previously shown to mediate endocytic trafficking of β1-integrin and TLR4 by directing endocytosed receptors to Rab11-positive recycling endosomes[33,60]. In these contexts, CD13 was internalized into endosomes with the receptors[33,60]. The role of CD13 in apical protein endocytic recycling pathway appears distinct from receptor-endocytic pathways because we did not observe overlap of CD13 with Rab11-containing apical recycling endosomes. Endosomal membranes are sub-compartmentalized with subsets of Rab11-containing recycling membranes located subapically which likely have multiple pathways for entry and exit[61,62]. Our data are consistent with a model by which CD13 at the cell periphery help to select cargo for delivery through a subset of Rab11-endosomes to the internal apical membrane. The physical connection between CD13 and Rab11-containing structures we observe likely reflects the presence of both in a connected subapical endosomal compartment[63]. In the absence of CD13, Par6 and likely other components of the Crumbs complex, are retained on the peripheral plasma membrane and excluded from Rab11-positive apical recycling endosomes that traffic to the AMIS. These endosomes may contain other apical cargo that are not regulated by CD13.

Rab35 is responsible for capturing vesicles by associating with podocalyxin to initiate internal apical specification[40]. In the absence of Rab35, these vesicles cannot dock and apical proteins are recycled to the cell periphery[44]. Our data reveal that CD13 precedes Rab35 accumulation at the AMIS and is required for

Rab35 localization. Therefore, in the absence of CD13, endosomes containing apical cargo not left behind at the cell periphery are unable to dock. Therefore, CD13 functions at multiple steps to control the delivery of apical proteins to the internal apical initiation site.

In summary, we propose the following model describing CD13 function in orienting apical-basal polarity. CD13 associates with the apical Par6/Crumbs complex (Fig. 8c, step 1) and couples it to the subapical endosomal compartment for Rab11-dependent delivered to the midbody to initiate apical membrane formation (Fig. 8c, step 2). In parallel, CD13 localizes to the newly formed cell-cell contact, and redistributes to the early AMIS (Fig. 8c, step 3), which is required to position Rab35 to capture incoming apical recycling endosomes (Fig. 8c, step 4).

Given the strong polarity effects we observe, it is interesting that CD13 knockout mice are viable[64–66]. Whether or not epithelial architecture and polarity is perturbed in these mice has not been characterized to our knowledge. However, genetic robustness to protect essential developmental processes through functional compensation is well documented[67]. For example, whole-body or tissue-specific deletion of other established polarity genes (e.g., Crb3, Prkci, Scrib) do not display gross loss of cell polarity or developmental defects that would be predicted for loss of polarity[68–70], supporting the idea of functional redundancy in polarity pathways. Like CD13, other M1-class aminopeptidases have enzyme-independent receptor functions[71], but whether these, or another protein, can compensate for CD13 to regulate polarity is not known.

In summary, our study identifies CD13 as an essential protein required for symmetry breaking by enabling redistribution of apical proteins to an internal position necessary for lumen formation.

## Methods

**Cell culture**. The Caco-2 human intestinal epithelial cell line was purchased from American Type Culture Collection (HTB-37). Caco-2 cells were cultured at 37 °C in 5% $CO_2$ in DMEM (Wisent #319-005-CL) supplemented with 10% fetal bovine serum (Wisent #080-150), 100 U/ml penicillin, 0.1 mg/ml streptomycin (Wisent #450201EL). For 3D culture, Caco-2 cells were seeded in 8-well μ-slides (Ibidi #80826) at a density of $1.25 \times 10^4$ cells per well on top of a thin layer of 100% GelTrex (ThermoFisher Scientific #A1413202) in media supplemented 2% GelTrex. After one or ten days in adherent culture, cells were collected for immunofluorescence. Human embryonic kidney cell line HEK293LT (ATCC), were cultured at 37 °C in 5% $CO_2$ in DMEM supplemented with 10% fetal bovine serum, 100 U/ml penicillin, 0.1 mg/ml streptomycin.

**DNA and shRNA constructs**. The pLX317-CD13-V5 plasmids were purchased from Sigma (TRCN, CD13 TRCN0000476147). pK-myc-Par6C was a gift from Ian Macara (Addgene plasmid # 15474). pWPI was a gift from Didier Trono (Addgene plasmid #12254). GFP-Rab35 WT was a gift from Peter McPherson (Addgene plasmid # 47424). GFP-Rab11 were provided by Robert Lodge. pSecTag_Myc-CRB3A was provided by Patrick Laprise. pWPI-myc-EGFP, pWPI-myc-aPKCι, pWPI-myc-Par6, pWPI-CD13-mCh, pWPI-EGFP-Par6, pWPI-flag-Par6wt were obtained by subcloning cDNA products to pWPI. pLX317-CD13$^{Y6F}$-V5, pLX317-CD13$^{Y6E}$-V5, pLX317-CD13$^{S8A}$-V5, pLX317-CD13$^{S8D}$-V5, pLX317-CD13$^{H388A}$-V5, pLX317-CD13$^{H392A}$-V5, pLX317-CD13$^{E411A}$-V5, pK-myc-Par6C$^{\Delta16-95}$, pK-myc-Par6C$^{\Delta95-168}$, pK-myc-Par6C$^{\Delta168-246}$, pWPI-flag-Par6$^{K19A}$, pWPI-flag-Par6$^{\Delta Pro136}$, and pWPI-flag-Par6$^{M235W}$ were generated using the QuikChange II site directed mutagenesis kit according to manufacturer's instructions (Agilent #200523).

shRNAs targeting human CD13 mRNA in pLKO were acquired from the McGill Platform for Cellular Perturbation (MPCP). Caco-2 cells were infected with lentiviral supernatants and selected by the addition of 20 μg/ml puromycin for 10 days. The shRNA sequences are available in Supplementary Table 1. Lentiviral supernatants were produced in HEK293LT cells 48 h after calcium phosphate co-transfection with packaging plasmid (psPAX2) and VSVG coat protein plasmid (pMD2.G). psPAX2 (Addgene plasmid # 12260) and pMD2.G (Addgene plasmid # 12259) were gifts from Didier Trono. Caco-2 cells were infected with lentiviral supernatants and selected by the addition of 20 μg/ml puromycin for 10 days.

**Transient transfection**. HEK293LT cells were seeded at $2 \times 10^6$ cells per well in 100 mm dishes and transfected with plasmids using Polyethylenimine (PEI) as per

manufacturer's instructions (Sigma # 408727). Caco-2 cells were seeded at $4 \times 10^4$ cells per well in 24 well and transfected with plasmids using Lipofectamine LTX as per manufacturer's instructions (Invitrogen #15338030). All experiments were performed 24 h post-transfection.

**Lentivirus production.** Lentivirus was produced using calcium phosphate transfection of HEK293LT cells in 15-cm dishes with 50 μg of lentiviral plasmid, 37.5 μg of packaging plasmid (psPAX2), and 15 μg of VSVG coat protein plasmid (pMD2.G). Viral supernatants were collected after 48 hrs and were concentrated by precipitation in 40% polyethylene glycol 8000 (Bioshop # PEG800.1) followed by centrifugation and then re-suspended in the culture medium. Concentrated virus was aliquoted and frozen at -80 °C, then an aliquot was titred in HEK293LT cells.

**Immunoblotting and immunoprecipitation.** Cells were lysed in RIPA lysis buffer (50 mM Tris-HCl, pH 8, 0.15 M NaCl, 0.1% SDS, 1% NP-40, 1% sodium deoxycholate, 50 mM NaF, 5 mM orthovanadate, 1 mM DTT) supplemented with proteinase inhibitor cocktail (Sigma # 11836170001). Total proteins were separated by SDS-PAGE and transferred to Nitrocellulose membrane (Bio-rad # 1620115). The primary antibodies used were: aPKCι 1/1000 (BD Transduction #610175), α-Tubulin 1/5000 (Sigma #T9026), Par3 1/1000 (Millipore #07-330); Par6B 1/1000 (Santa Cruz #sc-67393), V5 1/5000 (Thermo Fisher Scientific #R960-25); flag 1/1000 (Delta Biolabs # DB125), CD13 1/1000 (Abcam #ab108382), CD13 1/500 (Sigma #HPA004625), Pals1 1/1000 (Proteintech Group #17710-1-AP), myc 1/1000 (Origene #A150121), GFP 1/1000 (Abcam #ab13970), and Phosphotyrosine clone 4G10 1/1000 (Millipore #05-321). For immunoprecipitation, cells were washed twice with ice-cold PBS and then lysed in NP40 lysis buffer (150 mM NaCl, 1% NP-40, 50 mM Tris-HCl, 10 mM NaF, 1 mM NaVO₄, pH 8.0) containing a protease inhibitor cocktail and calyculin A. Lysates were precleared with Magna-Beads (Thermo Fisher Scientific #12321D) and then incubated with 2 μg of antibody or isotype control overnight at 4 °C. Antibodies were captured with MagnaBeads and washed three times with NP40 buffer.

**Immunostaining and imaging.** Cells from three-dimensional cultures were fixed with 2% paraformaldehyde/PBS for 10 min, permeabilized in 0.5% Triton X-100/10% Goat serum/10% fish gelatin/PBS for 1 h and incubated overnight in primary antibodies. The primary antibodies were used at the following dilutions: Par6B 1/200 (Santa Cruz #sc-67393), aPKCι 1/100 (BD Transduction #610175), Par3 1/200 (Millipore #07-330), CD13 1/100 (Abcam #ab108382), CD13 1/50 (Sigma #HPA004625), E-cadherin 1/200 (Cell Signaling #3195 S), ZO-1 1/100 (Cell Signaling #8193), Ezrin 1/200 (Cell Signaling #3145), Pals1 1/100 (proteintech group #17710-1-AP), β1-integrin 1/200 (Abcam #ab30394), V5 1/200 (Thermo Fisher Scientific # R960-25), laminin 1/200 (Abcam #ab11575), α-Tubulin 1/200 (Sigma #T9026), Phalloidin 1/100 (Invitrogen #A34055), and endoplasmic reticulum (ER) marker-Calreticulin 1/200 (Abcam #ab92516). The secondary antibodies conjugated to Alexa488-Donkey anti-Rabbit IgG (Jackson ImmunoResearch Laboratories #711-545-152), Goat Anti-Mouse IgG (Cy3) (Jackson ImmunoResearch Laboratories #115-165-166) and Alexa647-Donkey Anti-Mouse IgG (Jackson ImmunoResearch Laboratories #715-605-151) were used at 1:750. DNA was detected with Hoechst dye 33258. Confocal imaging was performed using LSM700 from Zeiss with 20×/0.8NA or 40×/1.4NA objective lenses and processed using FIJI/ImageJ software.

**Live-imaging.** Live-imaging of Caco-2 3D structures was performed using a LSM700 confocal microscope with environmental control (37 °C and 5% CO₂) using a LD plan-Neofluar 20×/0.4NA Korr M27 objective lens with 1% laser power and ZEN software (Zeiss). Caco-2 cells were virus-infected with CD13-mCherry, EGFP-Par6, or GFP-Rab11 in 2D culture for stable gene expression. Cells were seeded in 3D culture prior to performing live imaging. 8-bit images were captured every 25 min with 5 slices (28 μm) and two color channels.

**Temporal image correlation microscopy (tICM).** To measure protein dynamics, the tICM were implemented using a TIRF (total internal reflectance fluorescence) microscope. Caco-2 cells were transient transfected with CD13-mCherry and GFP-Rab11 in 2D culture. After 24 h, cells were seeded on 8 well Ibidi plate supplemented with 5% GelTrex-media at 37 °C in 5% CO₂ in DMEM supplemented with 10% fetal bovine serum, 100 U/ml penicillin, 0.1 mg/ml streptomycin for 24 h prior to performing live imaging. TIRF was obtained using a Spectral Diskovery unit (Spectral Applied Research, Richmond Hill, ON) attached to an inverted Leica DMI6000B microscope (Leica Microsystems, Wetzler, Germany) with a Leica Plan ApoChromat 63x/1. 47 NA TIRF oil immersion objective lens. The platform incorporates a 488 nm diode laser, 561 nm diode-pumped solid-state laser, 642 nm didode laser (Spectral Applied Research) with two ImageMX2 Digital EM-CCD Cameras (Hamamatsu, Hamamatsu City, Japan). A 100-Watt X-Cite 120 LED (370-700 nm) source was applied to allow visualization of fluorescence proteins by eye. The platform was integrated with MetaMorph 7.1 image acquisition software (Molecular Devices Inc.). Each image was set to collect 1000 frames with a 30 ms interval.

**Statistical analysis.** For analysis of the overlap of CD13 puncta with the endoplasmic reticulum (ER), local maxima were identified for CD13 puncta using the Find Maxima function in ImageJ, with output set to Maxima with Tolerance. Images of ER were thresholded using the Otsu method, and overlap was detected using Image Calculator AND function and analyze particles in ImageJ. The total number of CD13 puncta and the number that overlapped with ER were used to calculate proportion of CD13 puncta that overlap with ER.

To calculate cross-correlation between CD13 and Rab35, images were cropped to regions-of-interest containing CD13 and Rab35 puncta, which were then background subtracted so that background pixels had a value of 0 using ImageJ. Pearson's Linear Correlation Coefficient (ρ) was calculated on non-background pixels using MATLAB[72]. The p-value was calculated by running 1000 simulations of correlation where the pixel values from the CD13 channel were randomized. The p-value represents the proportion of simulations that had a higher correlation coefficient than the sample[72].

Comparison of two unpaired independent means were performed using a two-sided t-test. Data were assumed to be normally distributed. Statistics were analyzed using Excel, GraphPad Prism 6, JMP14, and MATLAB.

**Reporting summary.** Further information on research design is available in the Nature Research Reporting Summary linked to this article.

## Data availability
Measured data generated in this study are provided in the Source Data file. The image datasets generated and analyzed during the current study are available from the corresponding author on request. Source data are provided with this paper.

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

## Acknowledgements

This work was supported by a Canadian Institutes of Health Research grant (PJT-156271) to LM. LM is a Fonds Recherche du Québec – Santé Research Scholar.

## Author contributions

L.-T.W. performed all experimental studies. L.-T.W. and A.R. performed tICM experiments. L.-T.W. and L.M. carried out analysis. C.E.M. and L.M. supervised the work.

## Competing interests

The authors declare no competing interests.
