## [Peer Review File · Nature Communications]

REVIEWER COMMENTS

Reviewer #1 (Remarks to the Author):

Manuscript NCOMMS-20-33310

CD13 orients the apical-basal polarity axis necessary for lumen formation

Li-Ting Wang, Abira Rajah, Claire M Brown, Luke McCaffrey

From this study it is clear that overexpression or lack of CD13 influences epithelial polarity and lumen formation, and overall has interesting elements. However, in many cases the conclusions are not supported by the data, leaving the underlying mechanisms by which CD13 may promote these processes uncertain. Some discussion regarding the strong in vitro phenotype seen in in vitro 3d cultures seems while unchallenged CD13 KO mice are healthy and viable should be addressed (1-3). In my opinion the manuscript is not ready for publication in Nature Communications.

- Key results:

This study uses knockdown and overexpression of CD13 to demonstrate that this transmembrane molecule controls lumen formation by guiding known regulators of cell polarity to their correct positions in the cell to promote lumen formation. Numerous apical-basal proteins were mislocalized in cells engineered to lack CD13 and IP/Western blots were used to demonstrate that they were present in a common complex. Rescue experiments using various CD13 functional mutants showed that the phosphorylation site Y6 is necessary for this CD13 function.

- Validity: Does the manuscript have flaws which should prohibit its publication?

The major flaw in this study is the use of tagged CD13 constructs throughout which may or may not localize with endogenous CD13, thus bringing many experiments into question. Given these limitations and the likely pleiotropic actions or at least consequences of actions of CD13, many interpretations overstate what largely descriptive data actually reveal.

- Originality and significance:

The topic is not entirely novel: It has already been shown in a number of publications, (referenced in the manuscript) that CD13 is highly polarized in epithelial cells, a regulator of Rab 11+ endocytosis, localization of b1integrin and other proteins, endocytic trafficking (although not directly in lumen formation), and cytoskeletal rearrangements, all of which have been linked to lumen formation. More experiments and extra controls need to be performed to increase the impact and to support the conclusions and model of the authors.

- Of immediate interest?

The connection between CD13 with lumen formation has not been described and the work could be of interest in the field of developmental biology. I feel it would be better suited to a more specialized journal.

- Data & methodology:

While the approach is valid, the methodology in many cases the data do not support the conclusions and the experimental methods are not sufficiently detailed.

- Appropriate use of statistics and treatment of uncertainties:

The results are overinterpreted in many cases.

Some additional statistics are required as outlined in detailed review.

- Suggested improvements:

Verify that tagged proteins faithfully recapitulate endogenous localization throughout.

References: appropriately referenced.

- Clarity and context: Is the abstract clear, accessible?

Yes.

Specific Points:

Figure 1A, B, s1A-G. Endogenous CD13 and the tagged CD13-V5 clearly localize differently in cysts. Endogenous CD13 appears to extend from the apical surface, around the nuclei and is also found on the basal surface, while the V5 is limited to the apical surface. Tagging membrane proteins often affects proper localization which appears to be the case here. The claim that the antibody staining is non-specific is not convincing. Finally, the CD13-V5 cysts look quite different than endogenous, with rounder nuclei, more defined junctions and clearer apical/basal demarcations. Unfortunately, the remainder of the study is based on cells containing tagged, transfected CD13. These should be verified in non-transfected cells, perhaps using a different mAb to CD13.

Figure 1C-I – All western blots should be quantified and statistically analyzed to verify significance. The number of technical and biological repeats should be specified in each figure.

Figure 1C, D – these panels are confusing. In the text, the authors refer to Immunoprecipitation from CD13-V5 Caco2 cells, while the legend indicates HEK293 cells (apparently transfected with tagged vectors for EGFP, aPKC, Par6 and CD13?). Immunoprecipitation with myc-Par6 pulled down very little CD13 in 1C, but IP with V5-CD13 in 1D pulled down substantially more Par6. Why the difference?

Figure 1C-, a control blot for myc is needed to confirm pull down of the myc-tagged proteins.

Figure 1E – Need to show Par6 blot as confirmation of specific pull down.

Figure 1F – Again needs Par6 blot to confirm pull down. Also, CD13 blot to show that Par6 is pulling down CD13 in this assay.

Figure 1H – IP with myc Ab pulls down markedly more CD13 in WT myc-Par6 containing cells than in 1C. Are these the same HEK293 cells? CD13 pull down is reduced in cells containing the Crib and PDZ domain deletions. Also, a faint Pals1 band is clearly present in the last 2 lanes of IP as well.

Figure 2A – Is cell viability the same in the absence of CD13? Also, the figure title should be CD13 knockdown, rather than knockdown CD13.

Figure 2B – statistics?

Figure 2H – The presence of 50% cysts with a single lumen and 50% with internalized ezrin while CD13 loss is >85% suggests that the process may be more complex than mere presence or absence of CD13.

Figure 2J- appears to be a quantification of 2G and should be designated 2I for clarity.

Figure 3A – It should be clearly stated that these represent immunostains of fixed cells. The text describes the localization of CD13-mCh and refers to Fig 3A and B in the same sentence. It is confusing.

Figure 3B- The faithful localization of an additional set of tagged proteins for tracking protein dynamics in live cell imaging must be verified by co-localization with endogenous protein, particularly in view of the ambiguity introduced in Fig 1. Also, the correlation between panels B and C are not clear. Which frame does 3C t0.00 correspond to in panel 3B or which stage of lumen formation in A? Is the diffuse CD13-mCh distribution at t9:35 typical? It wasn't apparent in other experiments.

Figure 4 – The statements that “CD13 is required to direct apical membrane components to the midbody to initiate internal apical membrane formation” and “Depletion of CD13 disrupts early

polarization of 3D Caco-2 cysts by mislocalizing Par6 and the Midbody” are somewhat overinterpreted. While loss of CD13 clearly results in these phenotypes, the authors have not yet established that link with the experiments included in this Figure.

Figure 5 - Figure title should be CD13 knockdown.

Figure 5E – Which of the 2 bands in the input is Rab11? The IP with V5 is hard to determine if the pull down is Rab11 (strong IgG bands make it difficult to see Rab11). Blotting for Rab11 would help. Using Rab11 inhibitors or Rab11 KD in CD13 WT cells to mimic shCD13 phenotype would solidify this data.

Figure 6B – This is again very confusing. The untransfected controls show that Par6 localization is clearly dependent on endogenous CD13. However, it appears that overexpression of CD13-V5 generates two phenotypes: in the presence of endogenous CD13 (in shScr) cysts appear larger and V5 is apical but again, also between nuclei as well as basal, while in cells lacking endogenous CD13 (shCD13), V5 clearly is confined to the apical membrane.

Figure 6D - What is the percentage of rescue? Similar to 6C?

Figure 6E- Difficult to tell if Par6 is present in either with either of the two V5 IPs, although this was much clearer in the blot in 1D.

1. Rangel R, Sun Y, Guzman-Rojas L, Ozawa MG, Sun J, Giordano RJ, Van Pelt CS, Tinkey PT, Behringer RR, Sidman RL, Arap W, Pasqualini R. Impaired Angiogenesis in Aminopeptidase N-Null Mice. *Proc Natl Acad Sci U S A.* 2007;104(11):4588-4593.
2. Winnicka B, O'Connor C, Schacke W, Vernier K, Grant CL, Fenteany FH, Pereira FE, Liang B, Kaur A, Zhao R, Montrose DC, Rosenberg DW, Aguila HL, Shapiro LH. Cd13 Is Dispensable for Normal Hematopoiesis and Myeloid Cell Functions in the Mouse. *J Leukoc Biol.* 2010;88(2):347-359. 2908940
3. Zotz JS, Wolbing F, Lassnig C, Kauffmann M, Schulte U, Kolb A, Whitelaw B, Muller M, Biedermann T, Huber M. Cd13/Aminopeptidase N Is a Negative Regulator of Mast Cell Activation. *FASEB J.* 2016;30(6):2225-2235.

Reviewer #2 (Remarks to the Author):

The manuscript by Wang and colleagues examines the function of CD13 in apical-basal polarity establishment during lumen formation in 3-Dimensional culture of Caco-2 cells. This is an elegant analysis that identifies CD13 as a potential anchor for apical Par6 and positioning of Rab11 vesicles at the Apical Membrane Initiation Site (AMIS) during de novo apical polarisation and lumen formation. This is a clear and well-executed study.

Some clarification of statistical analyses and interpretation of data remain, which when addressed, will make an excellent contribution to the field of cell polarity.

Major comments:

Statistical and replicate information

There is an absence of quantitation in biochemical characterisation throughout. Please add replicate information and statistical analysis for immunoprecipitations. For instance, in Fig 1H-I, IP efficiency and input loading of exogenous protein need to be considered to evaluate binding efficiencies of truncated/mutated proteins. IPs are largely comprised of exogenous overexpression of tagged proteins. Do endogenous proteins interact? At least the key interactions could be validated.

In Fig 4, do dots represent independent experiments or acini quantified from a single experiment?

Rab11-CD13 cross talk

How does CD13 become concentrated at the AMIS? It appears from live imaging (Fig 3B) the initial cell-cell contact location of CD13-mCherry (t=0) transitions to a vesicular pool prior to concentration at the AMIS. What is the endocytic route for this? Is it Rab11a? From Fig 4A, the authors conclude that "CD13 accumulation at the future apical membrane site precedes Rab11." However, in Fig 4A, the vesicular stage of CD13 is missing in the presented staging. If CD13 precedes Rab11a, how does CD13 get to the AMIS? Is there an additional stage where CD13 is moving via Rab11 vesicles?

How is Rab11a so clearly IPing with CD13 while only (perhaps) small co-localisation is observed? Is this due to the co-overexpression? Can this association be detected with endogenous protein?

How exactly is Rab11 associating with CD13? Given that there is such a short cytoplasmic domain, does one of the mutants that the authors have used regulate Rab11 association?

Rab11a is stated as being 'at' the plasma membrane. How do the authors know that this is not (likely) just beneath?

CD13-Y6F immunolabelling has nuclear envelope and reticular localisation that is reminiscent of Endoplasmic Reticulum. What is the localisation of this mutant? As the authors report a lack binding of CD13-Y6F to Par6, but maintained Rab11 association, does the CD13-Y6F mutant disrupt Rab11 localisation?

Model and discussion

Missing from the model is how CD13 becomes accumulated at the AMIS. Could the authors speculate?

Why does a lack of CD13 cause accumulation of apical proteins at the cell periphery? Klinkert and Echard (Nat Comms 2016) showed that Rab35 has a similar function to the phenotypes proposed here, wherein loss of Rab35 causes apical protein to be internalised, but then without a central accumulation of apical proteins at the AMIS, internalised apical proteins trafficked back to the periphery. Does that also occur in CD13-depleted cells?

Minor comments:

Fig 1B. Also appears that CD13-V5 in puncta that sit below the apical Pals1 signal. Is this tight junction CD13?

Fig S1. CD13-V5 appears to bulge into the lumen in some images. Are these extracellular vesicles or apical membrane blebs in the lumen?

The authors state that 'Par6 did not colocalize with CD13 at the cell-cell adhesion stage, prior to apical membrane initiation, but was colocalized with CD13 at the AMIS and PAP stages (Fig. 3A).' There is clear colocalization of CD13 and Par6 at the cell periphery, but not the cell-cell contact.

I very much enjoyed reviewing this paper, which is a lovely piece of work.

-David M. Bryant, PhD

Reviewer #3 (Remarks to the Author):

This manuscript presents evidence that the apical membrane protein CD13/Aminopeptidase N has an important role in the establishment of apical/basal polarity in epithelial cells. While it is well established that CD13 is a component of apical membranes in most epithelia, this is the first demonstration that it has an active role in the biogenesis of apical membranes. The data clearly demonstrate that depletion of CD13 results in a failure to form apical lumina in 3D cultures of Caco-2 epithelial cells, and a redistribution of other apical proteins to the outer, ECM-facing surface of epithelial cysts. In this context, CD13 appears to associate with the Par6/Pals/Crumbs complex at the apical plasma membrane, through an interaction that requires Par6. Interestingly, CD13 accumulation at the nascent apical membrane appears to precede that of Rab11, a marker of apical recycling endosomes that deliver content to the forming apical membrane. CD13 has been shown to have important roles in other cell types that are independent of its canonical aminopeptidase activity, and the authors clearly demonstrate that this is also the case here; striking data show that apical lumen formation is strictly dependent on a tyrosine residue (Y6) in the cytoplasmic domain of CD13, although they do not define a function for it (see below).

In general, the data are clear and convincing, and largely support the conclusions drawn by the authors. Such an important role in apical membrane biogenesis is unexpected for a protein that has previously been thought of as merely a resident apical protein, and should be of interest to the general readership of Nature Communications. There are, however, several issues that need to be addressed:

1. All of the immunoblots need to be quantified and corresponding graphs shown. This is especially important since many of the interactions appear to be weak.
2. The significantly smaller size of CD13-depleted cysts suggests a reduced rate of growth, yet the authors claim that they observe no defects in abscission or multinucleated cells. Is the number of cells similar in CD13+ vs CD13-depleted cysts? Is the difference solely due to the fact that they don't form internal lumens?
3. Is the heterogeneity of the phenotypes observed in CD13-depleted cysts correlated with knockdown efficiency? Can this be determined by immunostaining for endogenous CD13?
4. The authors repeated refer to a role for Rab11 in endocytosis. What is the basis for this? Rab11 is primarily associated with recycling endosomes in non-polarized cells and with apical recycling endosomes in epithelia. To this reviewer's knowledge, it has no role in endocytosis per se, but rather in the redistribution of endocytosed cargo and delivery of that cargo to the apical membrane in epithelia. This point needs to be clarified.
5. The authors' statement that CD13 associates with Rab11 (top of p14) is a stretch. More likely a fraction of CD13 associates with Rab11-containing membranes (i.e. in endosomes), without requiring direct, or even indirect interaction. This should be clarified.
6. The CD13-Y6F mutant appears very diffusely localized in cells expressing it, and not associated with any peripheral membrane. Is it clear that this mutant escapes the ER after synthesis? ER retention seems plausible given that CD13 is a Type II membrane protein with a very short N-terminal cytoplasmic sequence. It is formally possible that the reason the Y6F mutant doesn't restore polarity in CD13-depleted cells is that it never gets out of the ER. This could be demonstrated clearly in 2D non-polarized cells that are easier to image.
7. Previous studies have shown that Y6 is phosphorylated by Src, and that this is important for its

non-catalytic function. Is Y6 phosphorylated in this context, and conversely, do Src inhibitors impair apical membrane biogenesis?

Response to Reviews

We sincerely appreciate the reviewers' comments and suggestions for improvements to our manuscript. We substantially revised the manuscript to address the concerns raised. Our point-by-point responses are described below in bold.

Reviewer 1:

1. From this study it is clear that overexpression or lack of CD13 influences epithelial polarity and lumen formation, and overall has interesting elements. However, in many cases the conclusions are not supported by the data, leaving the underlying mechanisms by which CD13 may promote these processes uncertain. Some discussion regarding the strong *in vitro* phenotype seen in *in vitro* 3d cultures seems while unchallenged CD13 KO mice are healthy and viable should be addressed (1-3). In my opinion the manuscript is not ready for publication in Nature Communications.

We thank the reviewer for their thorough examination of our manuscript for raising key issues. In our revised manuscript, we provided additional support for our conclusions on the role of CD13 in establishing apical polarity and lumen formation. These are described below in detail in response to specific points raised by the reviewer.

In regards to CD13 knockout mice, we acknowledge that CD13 knockout mice are viable and apparently healthy. However, to our knowledge, analysis of epithelial architecture and polarity has not been performed in published studies. Nonetheless, the simplest explanation is that of genetic robustness, in which functional compensation acts to ensure critical pathways during development, like cell polarity, can proceed when components of a pathway are disrupted. This is well described for polarity genes where some knockout mice retain polarity and can develop normally. For example, apical-basal polarity is retained in knockouts for established polarity genes *Crb3*, *Prkci*, *Scrib* (PMID: 26631503, 21744423, 26376988) Whether other M1-type aminopeptidases, or unrelated genes, can fulfil this function and compensate for CD13 during polarization *in vivo* is not yet known. We have addressed this topic in the revised Discussion.

2. The major flaw in this study is the use of tagged CD13 constructs throughout which may or may not localize with endogenous CD13, thus bringing many experiments into question. Verify that tagged proteins faithfully recapitulate endogenous localization throughout.

Figure 1A, B, s1A-G. Endogenous CD13 and the tagged CD13-V5 clearly localize differently in cysts. Endogenous CD13 appears to extend from the apical surface, around the nuclei and is also found on the basal surface, while the V5 is limited to the apical surface. Tagging membrane proteins often affects proper localization which appears to be the case here. The claim that the antibody staining is non-specific is not convincing. Finally, the CD13-V5 cysts look quite different than endogenous, with rounder nuclei, more defined junctions and clearer apical/basal demarcations. Unfortunately, the remainder of the study is based on cells containing tagged, transfected CD13. These should be verified in non-transfected cells, perhaps using a different

mAb to CD13.

Detecting endogenous CD13 has been an ongoing challenge. We have repeated immunostaining with a new anti-CD13 antibody that shows clear apical staining of endogenous protein in mature cysts, as well as during early stages of polarization (adhesion, AMIS, PAP, lumen). This is presented as new data in Figs. 1A, S1A, 2G, 3B, and 6C). The patterns observed are very similar to those observed with V5- or mCherry-tagged CD13 which supports that CD13 is localized to the apical membrane and is recruited early to sites of apical initiation. This is consistent with other reports demonstrating that CD13 expressed at the apical membrane in intestinal epithelium and Caco2 cells (PMID: 15919828, PMID: 20512993). Moreover, our functional studies using shRNA strongly support roles for CD13 at each of these stages including internalization of apical proteins from the cell periphery and establishing the apical initiation site (Figs. 2, 4A-C, 5B-F, 6C-D). Collectively this supports the reliability of our data with exogenous CD13 expression.

There is some heterogeneity in the structure of Caco2 cysts in terms of the thickness of the epithelial layer, with rounder nuclei present in thicker epithelial layers (for example see PMID 19001128), also our Fig. S4A and C. We do not observe consistent differences in nuclear roundness/epithelial thickness in cells expressing CD-V5 or non-transfected cells. We have replaced the image in Fig. 1B with one displaying a more representative shape/nuclear roundness. Our staining of junctions (E-cad and ZO1) and demarcation of apical/basal polarity is similar to other reports in Caco2 cells (PMIDs 19001128 and 21300793).

3. Figure 1C-I – All western blots should be quantified and statistically analyzed to verify significance. The number of technical and biological repeats should be specified in each figure.

We completely agree with this suggestion. Quantification has been added for the western blots using dot-plots to show experimental replicates with associated p-values. Technical and biological repeats have been added to the figure legends.

4. Figure 1C, D – these panels are confusing. In the text, the authors refer to Immunoprecipitation from CD13-V5 Caco2 cells, while the legend indicates HEK293 cells (apparently transfected with tagged vectors for EGFP, aPKC, Par6 and CD13?). Immunoprecipitation with myc-Par6 pulled down very little CD13 in 1C, but IP with V5-CD13 in 1D pulled down substantially more Par6. Why the difference?

We apologize for the confusion. Figure 1C shows that myc-Par6 (and myc-PKCi) can pull-down CD13-V5 in HEK293 Caco2 extracts, whereas Figure 1D (now Supplementary Fig. S1G) shows that CD13-V5 can pull-down endogenous Par6 (but not Par3) in Caco-2 lysates. It is difficult to make a direct comparison between the degree of pull-down between the two experiments since different cell lines, different tagged proteins, and different primary antibodies are used. Nonetheless, the purpose of these experiments is to demonstrate that CD13 and apical polarity proteins (i.e. Par6, aPKC, and Pals1) have the capacity to associate. Given their

overlapping localization at the apical membrane, we interpret this that they are part of a complex at the apical membrane.

5. Figure 1C-, a control blot for myc is needed to confirm pull down of the myc-tagged proteins.

This has been added to updated Fig. 1C.

6. Figure 1E – Need to show Par6 blot as confirmation of specific pull down.

This has been added to the figure (now Fig. 1D).

7. Figure 1F – Again needs Par6 blot to confirm pull down. Also, CD13 blot to show that Par6 is pulling down CD13 in this assay.

These blots have been added.

8. Figure 1H – IP with myc Ab pulls down markedly more CD13 in WT myc-Par6 containing cells than in 1C. Are these the same HEK293 cells? CD13 pull down is reduced in cells containing the Crib and PDZ domain deletions. Also, a faint Pals1 band is clearly present in the last 2 lanes of IP as well.

Fig. 1H is now Fig. 1G in the revised figures. These experiments used the same HEK293 cell line, however they are from different passages. Moreover, the Par6 plasmid backbones are different and the myc-IPs used different antibody lots, which likely contribute to some variation between experiments. Nonetheless, the experiments demonstrate that CD13 and Par6 can interact. We agree with the reviewer that CD13 is also reduced in semi-CRIB domain and PDZ deletion mutants. This is apparent with the blot quantification added. However, our data indicate that the PB1 domain of Par6 is essential for CD13 to associate. The semi-Crib and PDZ domains deletions also affect CD13 association, but to a lesser degree. Consistently, we also observed that these mutations also affect PKCi association, suggesting that the overall structure is required for efficient complex formation. Interestingly, the further from the PB1 domain the deletion is, the less severe the reduction in CD13 association. The domain point mutants retain associated CD13, suggesting that the overall structure of Par6 is likely important for associating with CD13 rather than through proteins associated with those domains. We do not yet understand how Par6 and CD13 interact, and whether it is direct or indirect. Unfortunately, experiments to determine if CD13 directly associates with Par6 using a peptide of the intracellular domain of CD13 to pull-down purified Par6 have been inconclusive. However, our data are consistent with a model by which CD13 associates with a Par6-containing polarity complex.

The reviewer is correct, there is residual Pals1 in the semi-CRIB and PDZ-deletion mutants. Quantification from multiple blots indicates that the reduction of Pals1 in semi-CRIB deletion

is minor. The reduced association of Pals1 with a Par6 PDZ-domain mutant is consistent with previous studies (PMID: 12545177).

9. Figure 2A – Is cell viability the same in the absence of CD13? Also, the figure title should be CD13 knockdown, rather than knockdown CD13.

Cell viability is reduced by CD13 knockdown as measured by cleaved caspase 3 staining. About 20% of spheroids had a single apoptotic cell and there was a small and statistically insignificant reduction in the mean number of cells/spheroid in CD13 knockdown. We have added this data to Fig S2A and B. We have corrected the figure title.

10. Figure 2B – statistics?

We have updated the figure to include the p-values.

11. Figure 2H – The presence of 50% cysts with a single lumen and 50% with internalized ezrin while CD13 loss is >85% suggests that the process may be more complex than mere presence or absence of CD13.

We agree with the reviewer here that it is likely more complex than a binary situation. As suggested by Reviewer 3, we examined endogenous CD13 in cyst with different phenotypes. We observe that in cysts with a prominent lumen, residual CD13 was detected at the apical membrane. Moreover, in the cells with mixed/intermediate phenotypes, we could detect low levels of residual CD13, whereas CD13 was not detected in spheroids with the most severe phenotype. In addition, we were unable to recover any clones for CD13 knockout cells using CRISPR/Cas9, indicating that loss of CD13 is not viable in this cell line. Therefore, the amount of CD13 present in individual cells can explain, in part, the variability of phenotypes observed. We have included this data in new Fig. 2G

12. Figure 2J- appears to be a quantification of 2G and should be designated 2I for clarity.

The reviewer is correct. The figure has been updated accordingly and panel I and J are quantification of panel H in the updated figure.

13. Figure 3A – It should be clearly stated that these represent immunostains of fixed cells. The text describes the localization of CD13-mCH and refers to Fig 3A and B in the same sentence. It is confusing.

We apologize for the confusion. We have now updated the sentence to:

After the first cell division, CD13-mCh transiently localized to the cell-cell contact surface (Fig. 3B), confirming our data from fixed images (Fig. 3A).

14. Figure 3B- The faithful localization of an additional set of tagged proteins for tracking protein dynamics in live cell imaging must be verified by co-localization with endogenous protein, particularly in view of the ambiguity introduced in Fig 1. Also, the correlation between panels B and C are not clear. Which frame does 3C t0.00 correspond to in panel 3B or which stage of lumen formation in A? Is the diffuse CD13-mCh distribution at t9:35 typical? It wasn't apparent in other experiments.

The diffuse CD13-mCh at t9:35 is the transition from CD13 being localized to the plasma membrane to coalescing at the AMIS, which we now call "Dispersion" in the figure. We have labeled intermediates as "Dispersion" and "Coalescence" to help compare stages between figure panels. We do typically see diffuse distribution of CD13 during AMIS formation. This can be difficult to observe because of saturating brightness of accumulated CD13 puncta in fixed images. Here we show a brightened image from Figure 1A, where low levels of diffuse CD13-V5 are visible surrounding the AMIS. Moreover, new Fig. 3B shows endogenous CD13 with diffuse puncta during AMIS formation.

Example image from Figure 3A showing normal and high-brightness version to visualize diffuse CD13 puncta surrounding the AMIS.

Immunostaining of endogenous CD13 requires conditions that do not preserve detection of direct fluorescence of mCherry. However, our new data showing endogenous CD13 (Figures 3B and 6C) localization patterns are consistent with our observations with mCherry.

15. Figure 4 – The statements that "CD13 is required to direct apical membrane components to the midbody to initiate internal apical membrane formation" and "Depletion of CD13 disrupts early polarization of 3D Caco-2 cysts by mislocalizing Par6 and the Midbody" are somewhat overinterpreted. While loss of CD13 clearly results in these phenotypes, the authors have not yet established that link with the experiments included in this Figure.

This is a fair point by the reviewer. We have revised the text to:

CD13 is required for accumulation of apical membrane components to the center of the midbody

and

Depletion of CD13 disrupts midbody position and apical protein accumulation to the midbody

16. Figure 5 - Figure title should be CD13 knockdown.

This has been updated.

17. Figure 5E – Which of the 2 bands in the input is Rab11? The IP with V5 is hard to determine if the pull down is Rab11 (strong IgG bands make it difficult to see Rab11). Blotting for Rab11 would help. Using Rab11 inhibitors or Rab11 KD in CD13 WT cells to mimic shCD13 phenotype would solidify this data.

The lower band is GFP-Rab11. We have included Rab11 blot to clarify interpretation of the data, well as quantification of the results. This is now in Fig. S5B.

We thank the reviewer for this excellent suggestion. To our knowledge, there are no Rab11-specific inhibitors, so we performed Rab11 knockdown. This is new data presented in Fig. 5G and shows that knockdown of Rab11 with 2 independent shRNA phenocopies depletion of CD13, with Par6 retained on the cell periphery at the 2-cell stage. We were unable to recover spheroids following longer periods in culture because Rab11 has a role in cytokinesis. Nonetheless, these new data demonstrate that Rab11 knockdown promotes inverted polarity like CD13 depletion and supports a functional link between CD13 and Rab11.

18. Figure 6B – This is again very confusing. The untransfected controls show that Par6 localization is clearly dependent on endogenous CD13. However, it appears that overexpression of CD13-V5 generates two phenotypes: in the presence of endogenous CD13 (in shScr) cysts appear larger and V5 is apical but again, also between nuclei as well as basal, while in cells lacking endogenous CD13 (shCD13), V5 clearly is confined to the apical membrane.

We apologize for the confusion with this figure. Caco2 cysts are heterogeneous in size (for example, Fig 2A). We attempted to choose images with cysts that are similar size for comparison in the figures. In this case, the cyst was somewhat larger in the control than the knockdown example. We have replaced this figure with a representative image similar in size to other structures. This is Fig. 7B in the updated manuscript.

19. Figure 6D - What is the percentage of rescue? Similar to 6C?

Wildtype CD13 fully rescues (100% of cells with internal Par6) and Y6F does not rescue (6%). The results show a similar effect as in Fig. 6C. We have updated the figure to include the quantification.

20. Figure 6E- Difficult to tell if Par6 is present in either with either of the two V5 IPs, although this was much clearer in the blot in 1D.

We have included quantification of all of our blots from multiple replicates to better represent the levels of proteins in the co-IPs. This blot and quantification are now in Fig. S6C.

Reviewer 2

The manuscript by Wang and colleagues examines the function of CD13 in apical-basal polarity establishment during lumen formation in 3-Dimensional culture of Caco-2 cells. This is an elegant analysis that identifies CD13 as a potential anchor for apical Par6 and positioning of Rab11 vesicles at the Apical Membrane Initiation Site (AMIS) during de novo apical polarisation and lumen formation. This is a clear and well-executed study.

Some clarification of statistical analyses and interpretation of data remain, which when addressed, will make an excellent contribution to the field of cell polarity.

We appreciate the positive comments by the reviewer.

Major comments:

1. Statistical and replicate information

There is an absence of quantitation in biochemical characterization throughout. Please add replicate information and statistical analysis for immunoprecipitations. For instance, in Fig 1H-I, IP efficiency and input loading of exogenous protein need to be considered to evaluate binding efficiencies of truncated/mutated proteins. IPs are largely comprised of exogenous overexpression of tagged proteins. Do endogenous proteins interact? At least the key interactions could be validated.

We have now included quantification of the blots from multiple experiments as dot-plots and mean intensity calculated for each. We have also included new data showing co-IP of endogenous CD13 with endogenous Par6 (Supplementary Figure S1). Moreover, as indicated in response to Reviewer 1, we provide new image data demonstrating endogenous CD13 localization at the apical membrane and during different stages of apical membrane initiation (Figs. 1A, S1A, 2G, 3B, and 6C).

2. In Fig 4, do dots represent independent experiments or acini quantified from a single experiment?

In Fig 4B and C, each dot represents the % of Internal Par6 from sample replicates, derived from two independent experiments. In Fig. 4F and I, each dot represents a single cyst collected from 3 independent experiments. We have updated figure legends to include information about replicates.

3. Rab11-CD13 cross talk

How does CD13 become concentrated at the AMIS? It appears from live imaging (Fig 3B) the initial cell-cell contact location of CD13-mCherry ($t=0$) transitions to a vesicular pool prior to concentration at the AMIS. What is the endocytic route for this? Is it Rab11a? From Fig 4A, the authors conclude that "CD13 accumulation at the future apical membrane site precedes Rab11." However, in Fig 4A, the vesicular stage of CD13 is missing in the presented staging. If CD13 precedes Rab11a, how does CD13 get to the AMIS? Is there an additional stage where CD13 is moving via Rab11 vesicles?

This is a very interesting set of questions. Indeed, CD13 appears to enter a vesicular pool from the adhesion, which we have called "Dispersion", (updated Fig. 3C) then the vesicles coalesce into the AMIS. We believe that CD13 accumulation at the emerging AMIS precedes Rab11 because we observe CD13 accumulation in the absence of overlapping Rab11. This is presented as a new Fig. 5B. We also performed object correlation and intensity profiles of CD13 and Rab11 to better visualize local peak intensities. These data show that Rab11-puncta are adjoining, but not overlapping with CD13 (new

Supplementary Figure S5A). Therefore, these data provide strong support that CD13 does not accumulate in Rab11 vesicles.

Since Rab35 also contribute to formation of the apical membrane by receiving apical cargo to the emerging AMIS, we also examined whether Rab35 may direct CD13 accumulation at the AMIS. This new data is presented in Fig. 6. We show that similar to CD13 (and unlike Rab11), Rab35 localizes to the adhesion membrane the accumulates at the AMIS and PAP as concentrated patches of Rab35. We find that CD13 localization becomes more restricted at the emerging AMIS before Rab35, but that they colocalize at later stages (i.e. PAP). However, we do not observe colocalization of CD13 with Rab35 in vesicles. To test if Rab35 was required to capture CD13-positive vesicles during AMIS formation, we knocked-down Rab35 and examined CD13 localization. CD13 was still able to accumulate at the emerging AMIS, however we did not observe compaction of the CD13 patch, consistent with a role for Rab35 in capturing apical vesicles to form a stable apical patch. In contrast, Rab35 did not localize properly in CD13-depleted cells. Together, our data support that CD13 accumulation precedes Rab11 and Rab35 to the AMIS and is one of the earliest events in apical specification. At present, we do not know the nature of the CD13 vesicles or how they accumulate but this will be of interest for future studies. We propose that there are multiple pools of CD13 (apical, vesicular, cell-adhesion) that that interconvert between pools/states during apical membrane initiation. Some proteins undergo basolateral-to-apical transcytosis that is independent of Rab11. One possibility is that the translocation of CD13 from the adhesion to AMIS progresses through a transcytosis-like process. We have updated the Discussion to include these points. Moreover, our data show that the CD13-Y6F mutant persists in the vesicular pool and does not translocate to the plasma membrane, which suggests that the intracellular domain may regulate the transition between the vesicular pool and other pools.

4. How is Rab11a so clearly IPing with CD13 while only (perhaps) small co-localisation is observed? Is this due to the co-overexpression? Can this association be detected with endogenous protein?

We acknowledge that co-IP as a methodology has limitations. For instance, the interaction of the proteins in a cell-lysate removes spatial organization and may over-estimate the interaction that occurs in intact cells. This may explain in part that Rab11 can clearly IP with CD13. Moreover, our IPs are performed on cell lysates from cells grown in 2D culture, which may affect the organization of the trafficking machinery. Thus, in isolation we interpret co-IP experiments as having the potential of proteins to associate. We made several attempts to detect the association with endogenous proteins, which were unsuccessful. To overcome these limitations, we included image cross-correlation as a biophysical means of evaluating proteins that are connected, in a spatial and temporal fashion in intact cells. This used TIRF microscopy and thus allows us to observe associations in living cells near the plasma membrane. Moreover, we see that the association is spatially restricted within the cells. Therefore, we think that the co-IPs likely overestimate the degree of association because they are free to interact in cell lysates. This data has been moved to Supplementary Fig. S5D-F in the updated manuscript.

5. How exactly is Rab11 associating with CD13? Given that there is such a short cytoplasmic domain, does one of the mutants that the authors have used regulate Rab11 association?

We tested the other CD13 mutants and they all show the capacity to associate with Rab11. Reviewer 3 suggested that CD13 likely does not directly associate with Rab11. Based on our revised colocalization data showing Rab11 adjoining, but not overlapping with CD13, and your comment below (6.) that Rab11a is not 'at' the plasma membrane, we do not believe that it is direct contact. Instead, CD13 appears to mediate the internalization and direction of selected cargo (i.e. Par6 and likely other components of the Crumbs complex) to Rab11-endosomes. As suggested by Reviewer 1, we provide new data showing that knockdown of Rab11 also leads to retention of Par6 at the peripheral membrane, providing a functional link between CD13 and Rab11 in mediating the early endocytic events of Par6. Our data are consistent with a model in which CD13 mediates coupling of apical proteins to Rab11-membranes within a connected endocytic network.

6. Rab11a is stated as being 'at' the plasma membrane. How do the authors know that this is not (likely) just beneath?

The reviewer raises a good point in line with published studies. Our data show Rab11a is adjoining the plasma membrane, but not overlapping. We have revised the text to read "near" the plasma membrane.

7. CD13-Y6F immunolabelling has nuclear envelope and reticular localisation that is reminiscent of Endoplasmic Reticulum. What is the localisation of this mutant? As the authors report a lack binding of CD13-Y6F to Par6, but maintained Rab11 association, does the CD13-Y6F mutant disrupt Rab11 localisation?

This is a good suggestion that was also raised by Reviewer 3, and one that we had not previously considered. We have examined the localization of CD13 with ER and indeed, there is partial overlap (about 20% of CD13-Y6F vesicles overlap with ER staining; this contrasts with 9% of CD13-WT vesicles), indicating that Y6F has somewhat impaired delivery from the ER. We present this new data in Fig. 7E and F.

Expression of CD13-Y6F alone does not affect Rab11 localization to the AMIS. This is consistent with our data that the Y6F mutant is in a vesicular compartment. Since vesicular CD13 is negative for Rab11, it is expected that it will not disrupt Rab11 localization. Since we see reduced association of CD13-Y6F with Par6, we wanted to determine if CD13-Y6F could associate other proteins to verify that it was not misfolded. Our interpretation is that CD13-Y6F is able to associate with Rab11-membranes in cell extracts, but in intact cells this interaction is excluded by the restricted localization of CD13-Y6F.

8. Model and discussion

Missing from the model is how CD13 becomes accumulated at the AMIS. Could the authors speculate?

We have updated the Discussion to provide some speculation in this area. We propose that CD13 exists in various pools (apical, vesicular, adhesion membrane) that interconvert during AMIS formation. While we do not know the mechanisms, it is reminiscent of basolateral-to-apical transcytosis observed by some proteins and can occur through Rab11-independent trafficking. Therefore, we think that cytokinesis triggers dispersion of CD13 from the adhesion into vesicles, which then coalesce to the early AMIS. This could involve other Rabs (like Rab5 in early endosomes), EEA1

(which is involved in apical CFTR transcytosis, PMID: 30975917) or other endocytic pathways not yet implicated in apical membrane formation. We have updated the Discussion to address these points.

9. Why does a lack of CD13 cause accumulation of apical proteins at the cell periphery? Klinkert and Echard (Nat Comms 2016) showed that Rab35 has a similar function to the phenotypes proposed here, wherein loss of Rab35 causes apical protein to be internalised, but then without a central accumulation of apical proteins at the AMIS, internalised apical proteins trafficked back to the periphery. Does that also occur in CD13-depleted cells?

Our observations indicate that apical proteins on the cell periphery largely stay there, and Rab11 vesicles traffic to the AMIS. These vesicles lack staining for Par6 and are likely devoid of the Crumbs complex, but they may contain other apical proteins. Based on our live imaging Rab11 patches appear at the centre, then move towards the periphery in multiple patches (Supplementary Fig. S5B), which is consistent with the Klinkert and Echard paper. As can be seen in Fig. 5C, Par6 is retained at the peripheral membrane. To obtain a clearer idea of Par6 localization during this process, we examined Caco-2 cells at the 2-cell stage where Rab11 was accumulating at the centre between nuclei. In these cases, we observe that Par6 is retained at the cell periphery. This new data is presented in Fig 5E-G. Our model is that Par6 and likely other members of the Crumbs complex remain at the cell periphery. Rab11 vesicles may contain other apical proteins traffic to the AMIS, but then disperse (i.e. Fig5C). Our new data showing that CD13 is required for Rab35 localization (Fig.6) provides a mechanism for how these vesicles would fail to dock. Therefore, retention at the periphery and recycling to the periphery may co-exist for different apical cargo.

10. Fig 1B. Also appears that CD13-V5 in puncta that sit below the apical Pals1 signal. Is this tight junction CD13?

We sometimes see these puncta staining for CD13 near the plasma membrane. However, co-staining with ZO1 indicates that they sit above tight junctions. See associated image below.

Caco2 cells immunostained for V5-CD13 (green) and ZO1 (magenta) show non-overlapping puncta.

11. Fig S2. CD13-V5 appears to bulge into the lumen in some images. Are these extracellular vesicles or apical membrane blebs in the lumen?

We sometimes observe these, which transiently bleb into the lumen then retract back into the plasma membrane without dissociating. Therefore, we think these are apical membrane blebs, rather than extracellular vesicles.

12. The authors state that 'Par6 did not colocalize with CD13 at the cell-cell adhesion stage, prior to apical membrane initiation, but was colocalized with CD13 at the AMIS and PAP stages (Fig. 3A).'

There is clear colocalization of CD13 and Par6 at the cell periphery, but not the cell-cell contact.

The reviewer is correct and we apologize that this was unclear. CD13 and Par6 colocalize at the cell-cell junction stage, but only at the cell periphery, not the cell adhesion, where CD13 localizes without Par6. We have revised this text to:

Par6 did not colocalize with CD13 at the cell-cell adhesion but they did colocalize at the periphery and at the apical site during stage AMIS and PAP stages (Fig. 3A).

Reviewer 3

This manuscript presents evidence that the apical membrane protein CD13/Aminopeptidase N has an important role in the establishment of apical/basal polarity in epithelial cells. While it is well established that CD13 is a component of apical membranes in most epithelia, this is the first demonstration that it has an active role in the biogenesis of apical membranes. The data clearly demonstrate that depletion of CD13 results in a failure to form apical lumina in 3D cultures of Caco-2 epithelial cells, and a redistribution of other apical proteins to the outer, ECM-facing surface of epithelial cysts. In this context, CD13 appears to associate with the Par6/Pals/Crumbs complex at the apical plasma membrane, through an interaction that requires Par6. Interestingly, CD13 accumulation at the nascent apical membrane appears to precede that of Rab11, a marker of apical recycling endosomes that deliver content to the forming apical membrane. CD13 has been shown to have important roles in other cell types that are independent of its canonical aminopeptidase activity, and the authors clearly demonstrate that this is also the case here; striking data show that apical lumen formation is strictly dependent on a tyrosine residue (Y6) in the cytoplasmic domain of CD13, although they do not define a function for it (see below).

In general, the data are clear and convincing, and largely support the conclusions drawn by the authors. Such an important role in apical membrane biogenesis is unexpected for a protein that has previously been thought of as merely a resident apical protein, and should be of interest to the general readership of Nature Communications. There are, however, several issues that need to be addressed:

1. All of the immunoblots need to be quantified and corresponding graphs shown. This is especially important since many of the interactions appear to be weak.

This issue was raised by all 3 reviewers and we agree it is important. In the revised manuscript we provide quantification and graphs for all immunoprecipitations and immunoblots.

2. The significantly smaller size of CD13-depleted cysts suggests a reduced rate of growth, yet the authors claim that they observe no defects in abscission or multinucleated cells. Is the number of cells similar in CD13+ vs CD13-depleted cysts? Is the difference solely due to the fact that they don't form internal lumens?

We have counted the number of cells/spheroids. While there is a small reduction in the mean number of cells per spheroid (1-3 cells fewer than control) this is variable and not statistically significant. Related to this, Reviewer 1 asked whether viability was different between CD13-deficient cells and controls. We do observe some apoptosis (1 cleaved-caspase positive cell in about 15-20% of CD13-deficient cysts, compared to about 2% of control cyst with an apoptotic cell), but this does not seem sufficient to dramatically alter the number of cells per spheroid. This new data is included in Supplementary Fig. S2A and B and has been addressed in the text. Moreover, the lumen accounts for 45-65% of the area of a cysts, and therefore loss of the lumen cavity can explain most of the reduction in size of Caco-2 cysts.

3. Is the heterogeneity of the phenotypes observed in CD13-depleted cysts correlated with knockdown efficiency? Can this be determined by immunostaining for endogenous CD13?

We thank the reviewer for excellent idea, which provides some clarity to the range of phenotypes observed. We were able to improve staining with a new CD13 antibody to detect endogenous protein. Indeed, CD13-knockdown cysts with a prominent lumen retain expression of CD13, those with an intermediate "mixed" phenotype with apical proteins on both a collapsed central lumen and periphery had low, but detectable CD13 at the internal apical surface, and cysts with inverted polarity have undetectable endogenous CD13. We have included this new data as Figure 2G.

4. The authors repeated refer to a role for Rab11 in endocytosis. What is the basis for this? Rab11 is primarily associated with recycling endosomes in non-polarized cells and with apical recycling endosomes in epithelia. To this reviewer's knowledge, it has no role in endocytosis *per se*, but rather in the redistribution of endocytosed cargo and delivery of that cargo to the apical membrane in epithelia. This point needs to be clarified.

We appreciate this point raised by the reviewer and agree with this comment. We have revised the text to clarify the known role for Rab11 in apical recycling endosomes, rather than endocytosis *per se*.

5. The authors' statement that CD13 associates with Rab11 (top of p14) is a stretch. More likely a fraction of CD13 associates with Rab11-containing membranes (i.e. in endosomes), without requiring direct, or even indirect interaction. This should be clarified.

We agree with the reviewer that CD13 unlikely directly interacts with Rab11. Related to this comment, Reviewer 2 also had issues with our description of Rab11 "at" the plasma membrane. Our revised model is that CD13 associates with the apical polarity complex, which is required for subsequent efficient loading to Rab11-apical recycling endosomes. As suggested by Reviewer 1, we show that knockdown of Rab11 leads to retention of Par6 at the apical membrane, providing a

functional link between CD13 and Rab11 in removal of Par6 from the peripheral membrane (new Fig. 5H). Based on our data showing the co-IP of CD13 with Rab11-containing membrane and cross-correlation TIRF data showing a physical connection between CD13 and Rab11 compartments and that the endosomal compartment is a connected tubular network with subcompartmentalization of Rab11 subdomains (PMID: 15126620, 23283983), our data are consistent with a model whereby CD13 is required to couple apical cargo to Rab11 membranes within the endosome compartment.

6. The CD13-Y6F mutant appears very diffusely localized in cells expressing it, and not associated with any peripheral membrane. Is it clear that this mutant escapes the ER after synthesis? ER retention seems plausible given that CD13 is a Type II membrane protein with a very short N-terminal cytoplasmic sequence. It is formally possible that the reason the Y6F mutant doesn't restore polarity in CD13-depleted cells is that it never gets out of the ER. This could be demonstrated clearly in 2D non-polarized cells that are easier to image.

We thank the reviewer for this suggestion, which we had not previously considered. Reviewer 2 also raised this point. We stained for ER and indeed, we see that CD13-Y6F partially overlaps with ER (20% of CD13-Y6F vesicles, compared to 9% of CD13-WT). Therefore, the Y6F mutant appears to be delayed in leaving the ER, however, most of the CD13 is in vesicles that fail to translocate it to the plasma membrane. We propose that CD13 exists in multiple pools (e.g. apical, vesicular, adhesion) and this indicates that Tyr6 is likely involved in regulating the distribution of CD13 between these pools. Future work will be needed to further understand how Y6 influences trafficking of CD13 to the plasma membrane.

7. Previous studies have shown that Y6 is phosphorylated by Src, and that this is important for its non-catalytic function. Is Y6 phosphorylated in this context, and conversely, do Src inhibitors impair apical membrane biogenesis?

These are interesting questions. We performed blots of CD13-WT and CD13Y6F using phospho-Tyrosine antibodies. However, we do not detect a difference in pTyr levels between these proteins. This data is included in Supplementary Figure S6C. Interestingly, we detect a pTyr band on both CD13-WT and CD13-Y6F. CD13 has 45 Tyr residues – all but one in the extracellular domain. Reports of secreted tyrosine kinases that can phosphorylate secreted and ER resident proteins on the extracellular surface (PMID: 25171405) raises the possibility that extracellular Tyr phosphorylation events may be important for CD13 function.

Treating cells with a Src-family inhibitor, Dasatinib, provides additional evidence against phosphorylation of CD13-Y6 in Caco-2 cells. The major effect seen when cells we treated with this inhibitor is collapse of the central lumen – an effect that likely has more to do with cortical tension than CD13-directed apical membrane protein trafficking.

Caco-2 cells treated with Src-family inhibitor Dasatinib during polarization and cyst growth.

REVIEWERS' COMMENTS

Reviewer #2 (Remarks to the Author):

OVERALL ASSESSMENT: It is my assessment that the reviewers have addressed all comments of Reviewer 1 and 2 with considered responses, new data, and updated discussion. As I have assessed the responses for both Review 1 and Reviewer 2, I provide those assessments individually below.

Reviewer 1 comments:

Summary of response to Reviewer 1 points: The authors have addressed all points raised by this review. The remaining query is to clarify how statistical significance is achieved in some experiments where only two experimental replicates are provided (Fig 5D, S2A S5C for Rab11, S6C). I support that presenting the fold change across two independent experiments is sufficient to draw a conclusion, but how significance was reached should be clarified.

Individual assessment of each of Review 1 Points:

Point 1. Discussion now added to address this point.

Point 2. With the addition of new antibody staining of endogenous CD13 the authors now demonstrate similar localisation of endogenous and tagged exogenous CD13.

The authors explanation of heterogeneity in thickness of Caco-2 cyst thickness being common in the literature is acceptable. The addition of similar cysts in Fig 1A-B is more appropriate.

Point 3. The addition of quantitation of western blotting is appropriate and highly supportive of the work.

Point 4. The author response is satisfactory (although a typo exists in the response calling Fig 1C 'HEK293 Caco2 extracts'. Since the figure legend states that these are HEK293, I am assuming the figure legend is correct. Perhaps the authors could indicate on the figure instances where Caco-2 or HEK203 are used.

Points 5-7. Appropriate addition of requested data.

Point 8. The reviewer queried differential association of mutants with binding partners. The addition of quantitation (also see point 3 above), now clarifies the banding patterns queried.

Point 9. Addressed with new data (Fig S2A, B).

Point 10. Addressed with addition of p-values.

Point 11. Variation in cyst phenotypes addressed with new data (Fig 2G), showing that type of polarity resulting from CD13 depletion is associated with different residual levels of CD13.

Point 12. Updated appropriately.

Point 13. Text updated appropriately.

Point 14. The updated labelling of live imaging of CD13-mCherry, and comparison to differential brightness of endogenous CD13 satisfactorily clarify the reviewer point.

Point 15. Appropriate softening of tone for previous overstatements.

Point 16. Updated appropriately.

Point 17. Addition of new data showing that knockdown of the CD13-interactor Rab11a gives a similar phenotype to CD13 depletion. Note that the figure references in the rebuttal letter are incorrect. These should be Fig S5C and Fig 5H.

Point 18. Updated with a more representative image (heterogeneity in this system is well known).

Point 19. Clarification provided in new Fig 7C. Note that all other plots except Fig 7C nicely show individual points for replicates. Please update Fig 7C to this format.

Point 20. Quantitation is provided, though it is unclear how the p-values were derived from only two experiments. This requires clarification.

Reviewer 2 comments:

Summary of response to Reviewer 1 points: The authors have satisfactorily addressed all points raised by this reviewer. This is a lovely manuscript.

Individual assessment of each of Review 1 Points:

Point 1. Addition of quantitation of biochemistry, IPs with endogenous protein, and new data with endogenous CD13 localisation addresses this point.

Point 2. Figure legends updated appropriately to clarify how quantitation is presented.

Point 3. The authors provide new data indicating that the vesicular transit of CD13 to the AMIS is not majorly via Rab11 or Rab35 vesicles, but that there is some dependence on these. While this still leaves an open question as to what transports CD13, this new data provides intriguing new avenues of research.

Point 4. An adequate response to the complexity of interpretation of immunoprecipitations.

Points 5-6. Although the mechanism of how Rab11 interacts with CD13 is not clarified, the authors have in lieu tested the functional relevance of Rab11a on CD13 by knockdown. This is satisfactory.

Point 7. The authors demonstrate that, as suggested, there is some retention of the CD13 mutant in the ER. This is now sufficiently accounted for in the manuscript.

Point 8. Discussion now adequately updated for model of how CD13 reaches the AMIS.

Point 9. New data addresses how Rab35 might be involved in the model proposed.

Points 10-12. Queries responded to adequately.

Reviewed by David Bryant

Reviewer #3 (Remarks to the Author):

The authors have done a good job of responding to previous reviewer concerns. Specifically, all blots are now quantitated and significant new data have been added that support and strengthen the authors' conclusions. Some over interpretations of the data have been tempered by changes to the text, and several confusing statements have been clarified. The discussion has been extensively modified to incorporate the new data and interpretations thereof. The revised manuscript is substantially improved over the original and is now appropriate for publication.

RESPONSE TO REVIEWER COMMENTS

We thank the reviewers for their comments. Our responses are in bold.

Reviewer #2 (Remarks to the Author):

OVERALL ASSESSMENT: It is my assessment that the reviewers have addressed all comments of Reviewer 1 and 2 with considered responses, new data, and updated discussion. As I have assessed the responses for both Review 1 and Reviewer 2, I provide those assessments individually below.

Reviewer 1 comments:

Summary of response to Reviewer 1 points: The authors have addressed all points raised by this review. The remaining query is to clarify how statistical significance is achieved in some experiments where only two experimental replicates are provided (Fig 5D, S2A S5C for Rab11, S6C). I support that presenting the fold change across two independent experiments is sufficient to draw a conclusion, but how significance was reached should be clarified.

We have removed significance tests from $n < 3$.

Individual assessment of each of Review 1 Points:

Point 1. Discussion now added to address this point.

Point 2. With the addition of new antibody staining of endogenous CD13 the authors now demonstrate similar localisation of endogenous and tagged exogenous CD13.

The authors explanation of heterogeneity in thickness of Caco-2 cyst thickness being common in the literature is acceptable. The addition of similar cysts in Fig 1A-B is more appropriate.

Point 3. The addition of quantitation of western blotting is appropriate and highly supportive of the work.

Point 4. The author response is satisfactory (although a typo exists in the response calling Fig 1C 'HEK293 Caco2 extracts'. Since the figure legend states that these are HEK293, I am assuming the figure legend is correct. Perhaps the authors could indicate on the figure instances where Caco-2 or HEK203 are used.

We apologize for this error, the figure legend is correct. We have updated figure 1 to indicate instances when Caco-2 or HEK293 cells were used.

Points 5-7. Appropriate addition of requested data.

Point 8. The reviewer queried differential association of mutants with binding partners. The addition of quantitation (also see point 3 above), now clarifies the banding patterns queried.

Point 9. Addressed with new data (Fig S2A, B).

Point 10. Addressed with addition of p-values.

Point 11. Variation in cyst phenotypes addressed with new data (Fig 2G), showing that type of polarity resulting from CD13 depletion is associated with different residual levels of CD13.

Point 12. Updated appropriately.

Point 13. Text updated appropriately.

Point 14. The updated labelling of live imaging of CD13-mCherry, and comparison to differential brightness of endogenous CD13 satisfactorily clarify the reviewer point.

Point 15. Appropriate softening of tone for previous overstatements.

Point 16. Updated appropriately.

Point 17. Addition of new data showing that knockdown of the CD13-interactor Rab11a gives a similar phenotype to CD13 depletion. Note that the figure references in the rebuttal letter are incorrect. These should be Fig S5C and Fig 5H.

We apologize for this error. We confirm that the figure references in the article are correct.

Point 18. Updated with a more representative image (heterogeneity in this system is well known).

Point 19. Clarification provided in new Fig 7C. Note that all other plots except Fig 7C nicely show individual points for replicates. Please update Fig 7C to this format.

We apologize for this oversight. We have updated Figure 7C (and Supplemental Figure 6b) to include individual points.

Point 20. Quantitation is provided, though is it unclear how the p-values were derived from only two experiments. This requires clarification.

We have removed significance tests from $n < 3$.

Reviewer 2 comments:

Summary of response to Reviewer 1 points: The authors have satisfactorily addressed all points raised by this reviewer. This is a lovely manuscript.

Individual assessment of each of Review 1 Points:

Point 1. Addition of quantitation of biochemistry, IPs with endogenous protein, and new data with endogenous CD13 localisation addresses this point.

Point 2. Figure legends updated appropriately to clarification how quantitation is presented.

Point 3. The authors provide new data indicating that the vesicular transit of CD13 to the AMIS is not majorly via Rab11 or Rab35 vesicles, but that there is some dependence on these. While this still leaves

an open question as to what transports CD13, this new data provides intriguing new avenues of research.

Point 4. An adequate response to the complexity of interpretation of immunoprecipitations.

Points 5-6. Although the mechanism of how Rab11 interacts with CD13 is not clarified, the authors have in lieu tested the functional relevance of Rab11a on CD13 by knockdown. This is satisfactory.

Point 7. The authors demonstrate that, as suggested, there is some retention of the CD13 mutant in the ER. This is now sufficiently accounted for in the manuscript.

Point 8. Discussion now adequately updated for model of how CD13 reaches the AMIS.

Point 9. New data addresses how Rab35 might be involved in the model proposed.

Points 10-12. Queries responded to adequately.

Reviewed by David Bryant

Reviewer #3 (Remarks to the Author):

The authors have done a good job of responding to previous reviewer concerns. Specifically, all blots are now quantitated and significant new data have been added that support and strengthen the authors' conclusions. Some over interpretations of the data have been tempered by changes to the text, and several confusing statements have been clarified. The discussion has been extensively modified to incorporate the new data and interpretations thereof. The revised manuscript is substantially improved over the original and is now appropriate for publication.